# Crisis of confidence averted: Impairment of exercise economy and performance in elite race walkers by ketogenic low carbohydrate, high fat (LCHF) diet is reproducible

Louise M. Burke[1,2]*, Avish P. Sharma[1,3], Ida A. Heikura[1,2], Sara F. Forbes[1], Melissa Holloway[1], Alannah K. A. McKay[1,4,5], Julia L. Bone[1,2], Jill J. Leckey[2], Marijke Welvaert[1,6], Megan L. Ross[1,2]

1 Australian Institute of Sport, Canberra, Australia, 2 Mary MacKillop Institute for Health Research, Australian Catholic University, Melbourne, Australia, 3 Griffith Sports Physiology and Performance, School of Allied Health Sciences, Griffith University, Gold Coast, Australia, 4 School of Human Sciences (Exercise and Sport Science), The University of Western Australia, Crawley, WA, Australia, 5 Western Australian Institute of Sport, Mt Claremont, WA, Australia, 6 University of Canberra Research Institute for Sport and Exercise, Canberra, Australia

* louise.burke@ausport.gov.au

**Data Availability Statement:** All relevant data are within the manuscript and its Supporting Information files.

## Abstract

### Introduction

We repeated our study of intensified training on a ketogenic low-carbohydrate (CHO), high-fat diet (LCHF) in world-class endurance athletes, with further investigation of a "carryover" effect on performance after restoring CHO availability in comparison to high or periodised CHO diets.

### Methods

After Baseline testing (10,000 m IAAF-sanctioned race, aerobic capacity and submaximal walking economy) elite male and female race walkers undertook 25 d supervised training and repeat testing (Adapt) on energy-matched diets: High CHO availability (8.6 g·kg$^{-1}$·d$^{-1}$ CHO, 2.1 g·kg$^{-1}$·d$^{-1}$ protein; 1.2 g·kg$^{-1}$·d$^{-1}$ fat) including CHO before/during/after workouts (HCHO, n = 8): similar macronutrient intake periodised within/between days to manipulate low and high CHO availability at various workouts (PCHO, n = 8); and LCHF (<50 g·d$^{-1}$ CHO; 78% energy as fat; 2.1 g·kg$^{-1}$·d$^{-1}$ protein; n = 10). After Adapt, all athletes resumed HCHO for 2.5 wk before a cohort (n = 19) completed a 20 km race.

### Results

All groups increased VO$_2$peak (ml·kg$^{-1}$·min$^{-1}$) at Adapt (p = 0.02, 95%CI: [0.35–2.74]). LCHF markedly increased whole-body fat oxidation (from 0.6 g·min$^{-1}$ to 1.3 g·min$^{-1}$), but also the oxygen cost of walking at race-relevant velocities. Differences in 10,000 m performance were clear and meaningful: HCHO improved by 4.8% or 134 s (95% CI: [207 to 62 s]; p < 0.001), with a trend for a faster time (2.2%, 61 s [-18 to +144 s]; p = 0.09) in PCHO.

**Funding:** This study was funded by a Program Grant from the Australian Catholic University Research Funds to Professor Louise Burke (ACURF, 2017000034).

**Competing interests:** The authors have declared that no competing interests exist

LCHF were slower by 2.3%, -86 s ([-18 to -144 s]; p < 0.001), with no evidence of superior "rebound" performance over 20 km after 2.5 wk of HCHO restoration and taper.

## Conclusion

Our previous findings of impaired exercise economy and performance of sustained high-intensity race walking following keto-adaptation in elite competitors were repeated. Furthermore, there was no detectable benefit from undertaking an LCHF intervention as a periodised strategy before a 2.5-wk race preparation/taper with high CHO availability.

## Trial registration

Australia New Zealand Clinical Trial Registry: ACTRN12619000794101.

## Introduction

Inadequate incentive and opportunity to replicate the results of small-scale research projects have contributed to a crisis of confidence in the reproducibility and transparency of many studies in the social and biomedical science literature [1, 2]. Strategies to combat this crisis include project registration, support for replication studies, declaration of conflicts of interest, sharing of data sets and full reporting of methodologies [2, 3]. A recent publication from our group, on the effects of adaptation to a ketogenic low carbohydrate, high fat (LCHF) diet on metabolism and performance in world-class endurance athletes [4], attracted keen interest from both social media and scientific circles. Given the high level of publicity around the project, we determined to undertake a replication study to test the robustness of our earlier findings, to address some of the challenges/criticism it received, and to extend both the scientific knowledge and practical implementation around its underpinning theme.

Our earlier study addressed claims from reviews [5,6] and anecdotal reports that adherence to a LCHF diet improves endurance performance by enabling an athlete's relatively unlimited body fat stores to better support substrate provision during exercise, preventing the performance decrements otherwise associated with depletion of the finite muscle carbohydrate (CHO) reserves [7]. Indeed, it has long been known that CHO restriction in concert with a high fat intake upregulates the release, transport, uptake and oxidation of fat in the muscle, even in endurance athletes whose training has already promoted such adaptations [8, 9]. Models have included chronic (> 2-wk) exposure to either a ketogenic (< 20 g·d$^{-1}$ CHO), high fat (80% of energy) diet [10] or a non-ketogenic restricted CHO (15–20% of energy), high fat (60–65% of energy) diet [11, 12] as well as short-term (5-d) adaptation to a high fat diet prior to strategies to restore CHO availability [13–15]. Despite remarkable increases in rates of fat oxidation during exercise of varying intensities, the translation to improved endurance performance has not been clearly demonstrated [9, 16]. Nevertheless, renewed enthusiasm in the application of a "keto" or LCHF diet (< 50 g·d$^{-1}$) CHO, moderate protein, high fat (75–80% of energy) for endurance sports [17] merited further inspection. At the time of undertaking our first study, the available literature on such a diet was limited to a single investigation of a 4-wk exposure in well-trained endurance athletes [10]. Accordingly, we determined the effects of 3 (.5)-wk adaptation to a LCHF diet during a period of intensified training on exercise metabolism and performance of world-class race walkers [4], comparing it with an energy-matched diet providing high CHO availability around all training sessions (HCHO). A third option

achieved periodisation of CHO availability (PCHO) with some training sessions being undertaken with high CHO intake to support training quality while others were undertaken with low CHO availability to enhance metabolic adaptation to the training stimulus [18]. This represents the modern approach of periodisation and personalisation of sports nutrition in which CHO availability is manipulated within and between days so that each exercise session can be undertaken with a dietary strategy that best supports the goals and characteristics of the workout [18]. Each walker was allocated to a dietary treatment according to their belief effect in the benefits of the intervention.

The training program was associated with a significant (~3%) increase in $VO_2peak$ with all dietary treatments [4], and in the case of the LCHF diet, markedly increased rates of whole-body fat oxidation during exercise. However, LCHF also increased the absolute oxygen ($O_2$) cost of race walking at velocities relevant to real-life race performance: $O_2$ uptake at a speed approximating 20 km race pace, expressed as % of new $VO_2peak$, was reduced in HCHO and PCHO but maintained at pre-intervention levels in LCHF. Performance of a 10,000 m track race was significantly improved for HCHO (6.6% faster); with a non-significant improvement of 5.3% for HCHO, and no change for the LCHF (non-significant 1.6% impairment). We concluded that adaptation to the LCHF diet negated performance benefits in elite endurance athletes, despite an increase in aerobic capacity, which we attributed in part to reduced exercise economy [4].

The intense interest in this paper (Attention Score of 939, representing the top 5% of all research outputs scored by Altmetrics at the time of submission of the current manuscript [19]) prompted an opportunity to scrutinise both the methodology and the results of this study with a view to undertaking a replication trial. A range of issues raised in published [20] and social media commentary about the first investigation is summarised in Table 1, with consideration of the merits of altering the methodology to allow greater scrutiny of the key findings of the earlier study. We determined that the goals of highest priority and feasibility were: (1) to replicate the 3.5 wk dietary interventions in a wider group of elite race walkers, including females, to investigate the robustness of the earlier findings of increased oxygen cost and impaired capacity for higher-intensity exercise (>75% $VO_2peak$) fueled by high rates of fat oxidation, and (2) to see if prior adaptation to a ketogenic LCHF achieves "carryover" benefits when participants resume a diet with high CHO availability for ~3 wk prior to competition. We hypothesised that our earlier observations of reduced exercise economy and performance impairments with the LCHF diet would be reproduced, and there would be no advantage to periodising adaptation to a LCHF prior to a return to training and competing with high CHO availability.

## Methods

### Ethical approval

This study, conforming to the *Declaration of Helsinki*, was approved by the Ethics Committee of the Australian Institute of Sport (AIS, #20161201) and registered with the Australian and New Zealand Clinical Trials Registry (ACTRN12619000794101). All participants provided informed consent after receiving comprehensive oral and written details of the protocol.

### Overview of study design

This study was conducted during a training camp (January-February 2017, which represented baseline preparation for the 2017 International Association of Athletics Federations (IAAF) race-walking season. The study took place at the Australian Institute of Sport (AIS) with participants living in athlete residences and undertaking all meals and training sessions under

**Table 1. Methodology of original study of LCHF diet in world class race walkers [4] for potential modification in replication study.**

| Feature of initial study | Discussion | Decision regarding replication study |
|---|---|---|
| Decision to allocate rather than randomly assign participants to treatment groups. Uneven matching of groups for baseline characteristics (aerobic capacity, performance best race times) | Participants were allocated to groups according to their belief/desire to undertake the treatment. Although this led to small but significant differences in baseline characteristics (LCHF group was higher in calibre/aerobic capacity than other groups), mixed modelling was used to account for this in the statistical analysis. | This methodology should be continued in the replication study since matching participants according to their beliefs will reduce the known performance bias due to placebo and nocebo effects [21]. In addition, the ability to receive a desired treatment is important in recruit world-class athletes to the study and ensuring maximum compliance. |
| Mix of repeated measures and parallel group design, and different conditions over two research camps (e.g., training group, environmental conditions on race day) | Study 1 was conducted over two separate training camps with participants attending 1 or 2 camps according to their availability. Smaller camp size allowed the research team to conduct study within their available resources and level of confidence around the control of diet and training. Mixed modelling was used to account for single or crossover participation in the study, as well different conditions between camps. | The experience gained from conducting a study of this magnitude provided the research team with confidence to manage an increased sample size in a single camp. Furthermore, publicity about Study 1 within the elite athlete community increased interest in participation, including the involvement of elite female race walkers. The increased statistical power associated with consolidating the project into a single research camp justifies the change in methodology. |
| Failure to provide an adequate version of the ketogenic LCHF diet [20] | Public criticism of the 'inadequacy' of the LCHF diet in original study was addressed [22], noting that it was constructed according to instructions and examples provided in a well-known book authored by key LCHF scientists [17] and that the criticisms reflect perceptions of the diet rather than a thorough understanding of food composition in relation to the diet restrictions. Menus deliberately maximised the inclusion of unprocessed/nutrient-rich foods within the strict macronutrient specifications of the LCHF and were consumed by participants under rigorous control. Full details of this methodology and dietary plans are available [23], including analysis that micronutrient content of diet typically met dietary reference intakes, although it was lower than that of the higher CHO diets due to fundamental restrictions on intake of many nutrient-rich foods [23] | The methodology around construction and implementation of diets [23] should be repeated in the replication study. |
| Inadequate duration of adaptation to LCHF [20] | The initial study was planned in view of pre-existing data that robust retooling of muscle to increase fat oxidation occurs in 5 d, with no further enhancement after this period [12], while initial fatigue and reduced exercise capacity is well reversed by 4 wk [10]. Furthermore, the well-known lay book written by the key proponents of LCHF identified that "if humans are given two or more weeks to adapt to a well-formulated low carbohydrate diet, they can deliver equal or better endurance performance compared to the best high carbohydrate diet strategy" [17]. Subsequently, it has been proposed that longer exposure is needed to gain true metabolic adaptation or to gain benefits from long-term exposure to high levels of circulating ketones [24–26]. However, much of this information is provided by social media commentary rather than rigorous examination of the time course of adaptation [25]. Although this warrants investigation key issues identified in the first study (e.g., change in oxygen cost of exercise associated with exceptionally high rates of fat oxidation due to established biochemical pathways) are unlikely to change over time. Furthermore, we note [4] that rates of exercise fat oxidation in study 1 are the highest recorded in the literature, including data from cross-sectional studies of endurance athletes who had been following self-selected ketogenic LCHF for $> 6$ mo [26, 27] | Although there is merit in investigating longer periods of adaptation to a ketogenic LCHF [25], the logistics involved with the current protocol involving full dietary/training control and the availability of world class athletes, present major challenges. These features are best aligned with the opportunity to replicate the findings of the first study and investigate the anecdotal reports of a "carryover" benefit to performance following the return to a CHO-rich diet (see below). As reviewed and shown by our earlier study [4], a training camp of this time period is well able to produce adaptations leading to performance changes. |

*(Continued)*

**Table 1.** (Continued)

| Feature of initial study | Discussion | Decision regarding replication study |
|---|---|---|
| Use of performance measure (10,000 m race) that is not relevant to endurance sport where glycogen depletion may become limiting | We designed a field protocol that maximised opportunities and incentives for participants to achieve a true measure of their performance (prize money, IAAF recognition), and that could be repeated ~ 3 wk apart. Although a 10,000 m race walk is not limited by glycogen depletion as might occur in the marathons/50 km race walk, it has other points of direct relevance to these longer races. Indeed, pacing strategies of the most successful race walkers in recent international 20- and 50-km event show pieces at significantly higher intensities at critical times within the race [28]. Furthermore, previous studies of specialist distance runners have found that 10-km personal best time is a good predictor of marathon performance [29] | The 10,000 m race walk event remains a logical and logistically suitable event to track endurance performance over the time course of the study and provides a good comparison to the 20 km road race (Australian National Championships) that will provide a convenient final performance metric following return to a diet with high CHO availability. It should be continued within the replication study. |
| Failure to note or encapsulate major performance improvements reported by some individual participants in the weeks following completion of the study | Commentary on lay [30] and social media coverage of the study noted that several participants achieved major performance improvements in the 2–3 wk after switching from LCHF to a diet with high CHO availability. It was suggested that LCHF for endurance athletes might have similarities to altitude training in which training under increased metabolic stress is associated with increased fatigue and impaired performance, but once the stress is removed and physiological characteristics of an exercise taper are integrated, there is a "carryover" effect that leads to a performance benefit [30]. | The "training camp effect" [31] whereby performance is enhanced simply due to an increased training load, high-level training partners and enhanced support/resources is well known; indeed other athletes in Study 1 on HCHO and PCHO diets also achieved performance improvements and "career breakthroughs" following Study 1. Nevertheless, the hypothesis that LCHF may create long-term adaptations that can be integrated with benefits of competing with high CHO availability is intriguing and offers a model of LCHF periodisation that can be practically investigated with the research opportunities of the replication study. |

supervision during the first two phases. The study was conducted using a parallel group design, with all participants completing a 5.5-wk structured dietary and training program, split into two phases (see Fig 1). Participants completed a Baseline testing block (4 d), before being allocated into one of three dietary groups according to their beliefs of performance enhancement. The first dietary intervention (Adapt) was undertaken over an intensified training program of 25 d inclusive of the Adapt testing block; this component of the study allowed a direct comparison with the results of our earlier study [4]. The second phase of the study (De-adapt) involved the change to a uniform dietary program and commencement of competition taper (10 d). A subgroup of participants then continued with an additional 7 d of self-chosen, but monitored, training and dietary preparation before competing in the Australian National 20 km race walking championships (Fig 1). The design and implementation of the study involved a pragmatic blend of rigorous scientific control and research methodology with real-world allowances needed to accommodate elite athlete populations.

**Participants and allocation to interventions.** Based on the experience from our first study [4], we established an *a priori* sample size of 10 participants per treatment group with the goal of having 8 participants complete each of the study components. We recruited highly competitive male and female race walkers, based on performances ranked by the IAAF, via word of mouth and targeted invitations from a key athlete and coach with whom the study was planned. Twenty-eight participants with international race experience were available, and as in the previous study, ranged from world class athletes to their high calibre training partners. During Wk1, one female race walker from the LCHF group was diagnosed with a chronic injury, while another male participant from the PCHO group incurred a severe upper respiratory tract infection and was unable to complete a significant number of the prescribed training sessions. These walkers were removed from the current analysis, leaving 26 data sets from

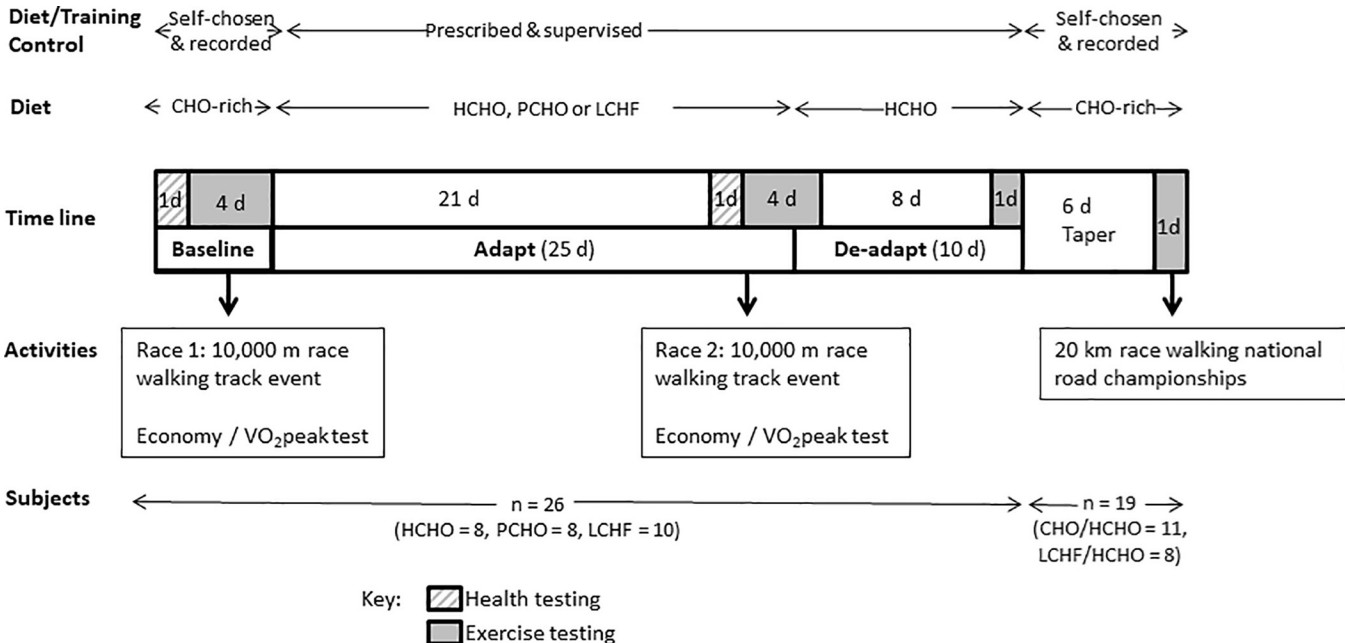

**Fig 1. Overview of study design undertaken by 26 elite race walkers including repetition of a previous intervention study involving 25 d adaptation (Adapt) to diets of high CHO availability (HCHO), periodised carbohydrate availability (PCHO) and ketogenic low CHO high fat (LCHF) [4] followed by 10 d return to high CHO diet (De-Adapt).** A cohort of 19 participants undertook a taper before completing a 20 km race walking event.

participants who completed the requirements of the 5.5-wk training study. Twenty of these participants were permitted by their national organizations to compete in the Australian 20 km race walking road championships and continued to contribute to the study with an additional week of tapered training and race preparation. One male race walker from the LCHF group was disqualified during the race due to a technical infringement and was not permitted to complete the course; therefore, 19 data sets were collected from this third phase of the study. The baseline characteristics of participants who participated at each stage of the study are summarised in Fig 2 according to CONSORT principles.

The Adapt phase (Fig 1) involved three different approaches to dietary support for the intensified training program: high CHO availability (HCHO), periodised CHO availability (PCHO) and low CHO, high-fat (LCHF). We repeated the protocol used in the previous study [4] of weighting the allocation of dietary treatments during the Adapt phase towards participants' belief systems in accordance with our recognition of the importance of the placebo effect. Prior to their arrival to the study camps, participants were educated about the results of the previous study and background to the current theme, including the potential for a carryover benefit from prior adaptation to the LCHF, and asked to nominate their preference for, or non-acceptance of, each of these interventions. We were able to allocate each of the race walkers to a preferred treatment for the Adapt phase of the study camp, while achieving reasonable matching of groups based on age, sex, body mass, current aerobic capacity and personal best for the 20 km race walking event (Fig 2), as well as their availability to compete in the 20 km race walking championships. Nevertheless, since many higher ranked 50 km race specialists chose the LCHF diet, residual baseline differences between groups were noted and included in the statistical analyses.

During the De-Adapt phase of the study, all participants were placed on a HCHO diet for 10 d (See Figs 1 and 2), merging the group into two cohorts: those who had been previously adapted to the ketogenic diet (= LCHF/HCHO) and those who had consumed CHO-rich diets

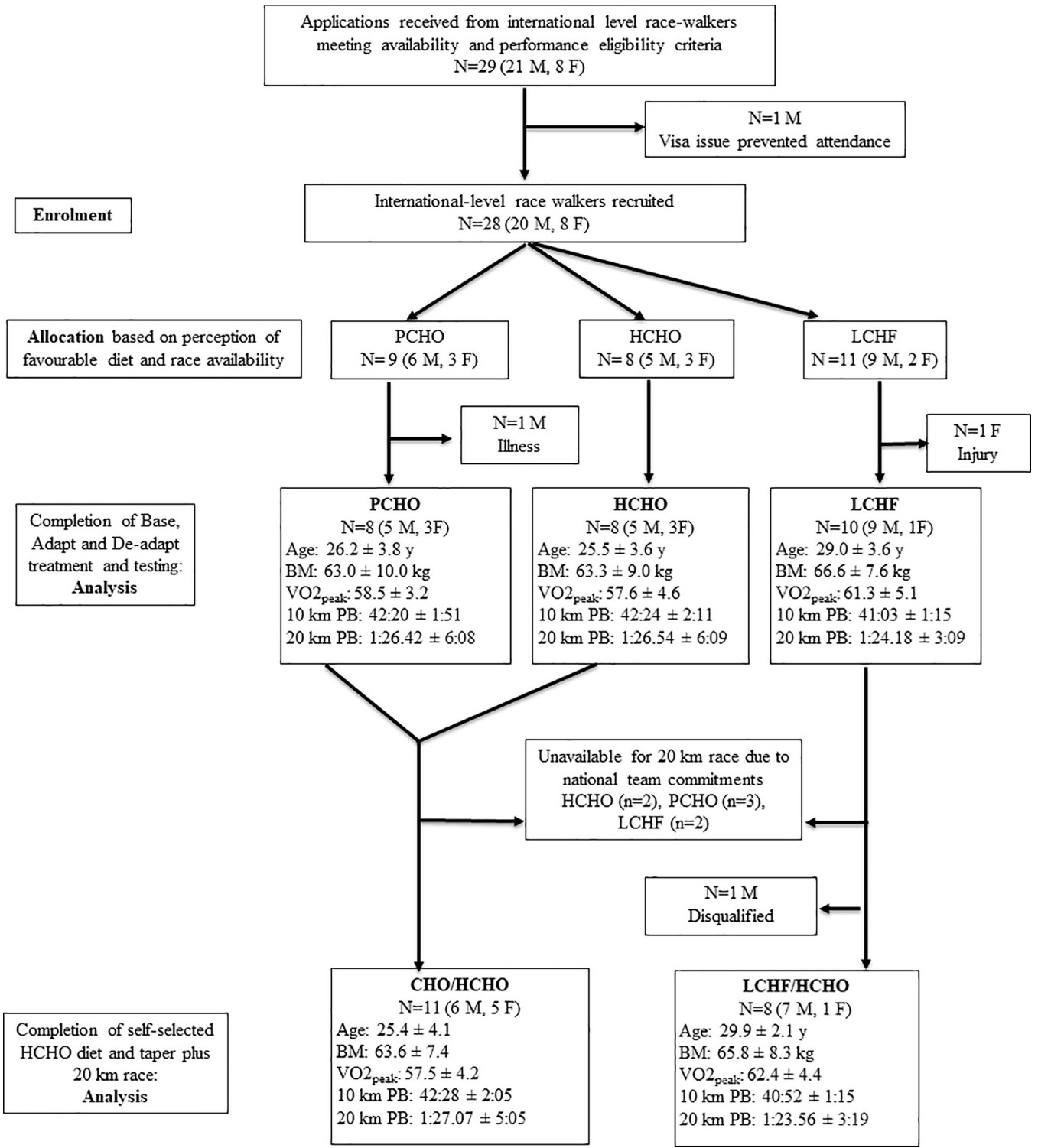

**Fig 2. Baseline characteristics of elite race walkers involved in diet-exercise study, noting participation in various phases of the protocol.** Characteristics: age (y), BM = body mass (kg), $VO_2$peak = maximal oxygen consumption during race walking test (ml·kg$^{-1}$·min$^{-1}$), 10 km PB = personal best in 10 km race walk (min:s), 20 km PB = personal best in 20 km race walk (h:min.s), where qualifying standard for 20 km event at 2016 Summer Olympic Games was 84 min for males and 96 min for females. CHO = carbohydrate, HCHO = high CHO availability, PCHO = periodised CHO availability and LCHF = low-CHO high-fat.

(HCHO or PCHO) throughout the study (= CHO/HCHO). Participants who were to continue to complete the 20 km race were then able to choose their own CHO-focused dietary plan, while recording all intake of foods and fluids for the 6 d of the Taper week, as well as their pre-race meal. Race intake (fluids and foods taken from the feed zone at each 2 km of the race) was self-chosen and recorded by the research team.

## Training intervention

Between the Baseline and Adapt testing protocols, all participants undertook 3 wk of intensified training incorporating race walking, resistance training and cross-training (e.g., running, cycling or swimming). Training sessions were undertaken as a group and were monitored by the research team as well as recorded by participants in a daily log. The training program model was repeated from our previous study [4], and represented collaboration with a world class race walker and coach to integrate the typical intensified training practices of competitive walkers with opportunities to implement the desired dietary interventions. Table 2 details the weekly program, noting six mandatory sessions that were undertaken under standardised conditions with external monitoring. The remaining sessions, undertaken according to the preference of the individual athlete, were recorded. Following Adapt testing, participants completed another 10 d of training and supervised diet, following the general theme of the previous program but commencing a competition taper by reducing the length of the twice-weekly long walks as well as self-chosen sessions to reduce overall volume. The final week of training for the cohort who competed in the 20 km race was self-chosen and recorded in training logs

## Dietary intervention

All foods and fluids consumed during the 5.5-wk of the Adapt and De-adapt phases of the study were provided and recorded by the AIS Sports Nutrition team. Menu construction and the preparation of meals and snacks were undertaken by professional chefs, food service

**Table 2. Overview of weekly training-diet intervention involving high carbohydrate (CHO) availability (HCHO), periodised CHO availability (PCHO) or low CHO high fat (LCHF) diets during 25 d Adapt phase of study in elite race walkers (n = 26).**

| Day | Diet | Monday | Tuesday | Wednesday | Thursday | Friday | Saturday | Sunday |
|---|---|---|---|---|---|---|---|---|
| **AM Training** | | **Easy 10 km** | **Easy 10–15 km*Gym** | **Long* (20–40 km)** | **Easy 10–15 km*Gym** | **Hill session*** | **Long*(25–40 km)** | **Easy 10km/ nil** |
| Dietary treatment around training session | HCHO (n = 8) | CHO pre, during, post | CHO pre, during, post | CHO pre, during, post | CHO pre, during, post | CHO pre, during, post | CHO pre, during, post | CHO pre, during, post |
| | PCHO (n = 8) | Fasted training; CHO post | Fasted training + low glycogen (Train low#); CHO post | CHO pre, during, post | Fasted training; CHO post | CHO pre + during; Nil CHO post (Recover low#) | CHO pre, during, post | Fasted training; CHO post |
| | LCHF (n = 10) | Minimal CHO; high fat | Minimal CHO; high fat | Minimal CHO; high fat | Minimal CHO; high fat | Minimal CHO; high fat | Minimal CHO; high fat | Minimal CHO; high fat |
| **PM Training** | | **Interval session*** | **Easy 10 km** | **Easy 10 km/nil** | **10–15 km** | **Easy 10–15 km*** | **Easy 10 km/ nil** | **Easy/nil** |
| Dietary treatment around training session | HCHO (n = 8) | CHO pre, during, post | CHO pre, during, post | CHO pre, during, post | CHO pre, during, post | CHO pre, during, post | CHO pre, during, post | CHO pre, during, post |
| | PCHO (n = 8) | CHO pre, during; minimal CHO post (Sleep low#) | CHO pre, during, post | Fasted training CHO post | CHO pre, during, post | Fasted training + low glycogen (Train low#); CHO post | CHO pre, during, post | CHO pre, during, post |
| | LCHF (n = 10) | Minimal CHO; high fat | Minimal CHO; high fat | Minimal CHO; high fat | Minimal CHO; high fat | Minimal CHO; high fat | Minimal CHO; high fat | Minimal CHO; high fat |

*Compulsory key training session #CHO periodization strategy [18].

dietitians and sports dietitians/nutritionists. Meal plans were individually developed for each athlete to integrate personal food preferences and nutrition requirements within her/his allocated dietary treatment. Meals were eaten in a separate dining area in a group setting with individualised meals being served for each athlete according to their meal plans. During each meal service, the weight of each food item was recorded using calibrated scales (accurate to 2 g). Individualised snacks were provided for intake between meals and before/during training sessions, with the requirement for their consumption to be cross-checked at the next meal. A range of "free foods and drinks" (foods with low energy such as fruits and vegetables, tea/coffee, water and artificially sweetened beverages) were provided in the participants' living area with a checklist to allow each participant to report on his day's intake at the first meal of the following day. Nutrition support during longer training sessions and after key sessions was provided at the training site by members of the research team and intake was recorded. Full details of this approach and sample menu plans are provided elsewhere [23]. Compliance to the dietary prescription and reporting requirements was checked daily.

The three dietary treatments implemented in the Adapt phase of the study, repeated from our earlier work [4], are summarised below and in further detail in Table 2. During the De-adapt phase, all participants followed the diet with high CHO availability.

1. High CHO availability (HCHO): Overall macronutrient composition 60–65% of energy from CHO, 15–20% protein, 20% fat; Similar daily CHO intake, with CHO consumed before, during and after training sessions. The diet represents sports nutrition guidelines from 1990s [32].

2. Periodised CHO availability (PCHO): Same overall macronutrient composition as HCHO, but spread differently between and within days according to fuel needs of training as well as an integration of some training sessions with high CHO availability (high muscle glycogen, CHO feeding during session) and others with low CHO availability (low pre-exercise glycogen, overnight fasted or delayed post-session refuelling). This strategy represents current guidelines [33, 34] and evolving evidence around benefits of strategic training or recovery with low CHO availability [35,36], at least in sub-elite athletes.

3. Low- CHO, high-fat diet (LCHF): 75–80% fat, 15–20% protein, $<50$ g·d$^{-1}$ CHO. A ketogenic diet following the guidelines previously reported [17] and investigated in a study undertaken with endurance cyclists [10].

Since the primary goal of the study was to evaluate these dietary treatments without any interference due to large or different changes in body composition, daily energy intake for each athlete was set at an energy availability of ~40 kcal·kg$^{-1}$ lean BM. A loss of fat mass of ~ 1 kg over the 25-d of the Adapt intervention was permitted and each athlete could request additional food at meals or from designated snacks and "free snacks" according to hunger, increases in training load from the baseline information or large fluctuations in BM above that expected with the glycogen/fluid shifts associated with the LCHF. When such variations occurred, they were achieved by following the macronutrient composition of the treatment diet and noted in the actual food consumed. During the De-adapt phase, where all participants followed the HCHO diet, further adjustment in energy intake allowed for reductions in training volume in consideration of the commencement of a race taper.

## Test block

Participants completed a 4-d testing block before (Baseline) and after (Adapt) the first 3 wk of the diet/training intervention (Fig 1). Components of the testing included laboratory testing

for aerobic capacity and economy, a 10,000 m race walking track race, a hybrid session of laboratory and field exercise and a hill training session. The current paper describes the outcomes of the first two tests; although the race was undertaken on the same day by all athletes, participation in the other testing sessions was rotated with the order and conditions being standardised for each participant.

**VO$_2$peak and walking economy.** Participants completed a treadmill test to assess economy and VO$_2$peak while race walking. This test was undertaken in an overnight fasted, rested state. The test was performed on a custom built, motorised treadmill (Australian Institute of Sport, Canberra, Australia). The test was comprised of four submaximal stages (for determination of submaximal VO$_2$ and walking economy). Each stage, three minutes in length, was immediately followed by an incremental ramp to exhaustion for determination of VO$_2$peak. As such, the total test duration was approximately 13 to 20 min, depending on a participant's time to exhaustion (TTE). The treadmill velocity for the first stage was dependent on each participants' most recent 10 km race time (9–12 km·h$^{-1}$), at 0% gradient, with the velocity being increased by 1 km·h$^{-1}$ with each subsequent stage. After each stage, a small (5 μL) capillary blood sample was taken from the fingertip for measurement of blood metabolites (see below), and Ratings of Perceived Exertion (6–20, Borg Scale was also collected). Heart rate (HR) was measured continuously during the test (Polar Electro, Kempele, Finland). Immediately following the completion of the fourth submaximal stage (approximately equivalent to 20 km race walk speed), the gradient of the treadmill was increased by 0.5 degrees every 30 s, until the participant reached volitional exhaustion. Expired gas was collected and analyzed using a custom built indirect calorimetry system described previously [37], with the final 60 s of gas collection accepted as steady state and used to calculate RER and O$_2$ uptake during submaximal stages. VO$_2$peak was taken as the highest 30 s value taken during the incremental portion of the test.

**Blood metabolites.** Capillary blood samples were used for this portion of the study to allow standardised collection of samples in both laboratory and field conditions where metabolites were to be assessed over the entire 5.5-wk intervention. Fingertip samples were collected and immediately processed for measurement of blood lactate (Lactate Pro 2, Akray, Japan), ß-hydroxybutyrate (FreeStyle Optium Neo, Abbott Diabetes Care, Victoria, Australia) and glucose (FreeStyle Optium Neo, Abbott Diabetes Care, Victoria, Australia) concentrations. To counter for any individual differences in the accuracy of these portable analyzers, each participant was assigned to a specific device for the duration of their involvement in the study.

**Calculation of respiratory exchange ratio (RER) and substrate oxidation data.** RER was calculated from steady-state expired gases collected over 1-min periods using the economy test and maximal aerobic capacity protocol. Rates of whole body CHO and fat oxidation (g·min$^{-1}$) were calculated for the second and fourth stage of the economy test from VCO$_2$ and VO$_2$ values using non-protein RER values [38]. These equations are based on the premise that VO$_2$ and VCO$_2$ accurately reflect tissue O$_2$ consumption and CO$_2$ production, and that indirect calorimetry is a valid method for quantifying rates of substrate oxidation in well-trained participants during strenuous exercise of up to ~85% of VO$_{2\ peak}$ [39]. We did not correct our calculations for the contribution of ketone oxidation to substrate use in the LCHF trials as we wished to directly compare our findings to our previous work [4] and other recent reports of substrate utilisation in ultra-endurance athletes who chronically consume LCHF diets [26, 27] where this error in the use of conventional equations to calculate fat and CHO oxidation from gas exchange information [40] has been accepted.

**10,000 m race.** Race walkers competed in IAAF sanctioned 10,000 m track races held on a 400 m outdoor synthetic athletics track (Canberra, ACT, Australia). To provide incentive for a maximal effort, prize money was awarded to athletes who achieved the highest percentage of their 20 km walking personal best when the times of the two races (Baseline and Adapt) were

combined. However, to ensure that the incentive was equally perceived by the participants, an allowance for the difference in post-diet performance observed in the previous study between the PCHO/HCHO groups and the LCHF group was applied. The formula for the correction (par time calculated from IAAF point score [41] for 10 or 20 km personal best time, with a 5% allowance provided to LCHF participants) was developed and implemented by a committee of the race walkers to promote confidence that each participant was optimally motivated to race as hard as possible during each race.

Each race commenced at 0830 and was conducted under IAAF rules which involved officiating by technical judges, invitation for participation by competitors external to the study, a feed zone allowing water intake on the outside lanes of the track in hot conditions, and photo finish electronic timing. Photo finish timing was used to provide official race times. Capillary blood samples were collected immediately before the start of the race and as each competitor completed the 10,000 m distance.

For the Baseline race, participants were permitted to consume their habitual pre-race diet for the 24 h prior to the race and their pre-race (morning) meal provided it was documented accurately. Use of performance supplements (e.g., caffeine) was discussed with each participant prior to the first race; permission was provided when it didn't interfere with the treatment diet, was documented, and was repeated for the Adapt race. For the Adapt race, participants followed the diet consistent with their treatment for the 24 h pre-race period and consumed a pre-race meal according to their dietary intervention (e.g., high in CHO or high in fat).

**20 km race.** The Oceania and Australian National 20 km race walking championships were conducted in Adelaide, South Australia on February 19, 2017 according to IAAF event rules previously described. The course consisted of a 2-km loop on which a "feed zone" was set up to allow competitors to receive race nutrition supplies (water, sports drinks, sports foods etc.) as well as an additional table on which water was supplied, principally as a cooling aid (e.g., to tip over the head and body). Participants were requested to record their pre-race meal and use of any performance supplements on the day of the race on food diaries supplied by the study, with the aid of portable scales, where practical. They were required to bring their own race nutrition supplies to members of the study research team stationed at the feed zone, from where they were handled according to IAAF rules. No outside nutrition support was permitted. Members of the research team collected information about participants' intake of fluid and carbohydrate during the race, using portable scales to monitor the change in mass of drink bottles of identified content over the course of the race, together with the observed intake of sports gels and confectionery and the pre- and post-measurement of any packaging around these products. Performances were obtained from the official race results and were compared between groups and against race times for the 10,000 m track event in two ways: by data normalisation (noting that a 20 km race is twice the distance of a 10,000 m race) and by converting each of the race times to its equivalent IAAF point score [41].

### Statistical analysis

Statistical analyses for metabolic and performance data were carried out using a General Linear Mixed Model using the R package lme4 [42, 43] allowing for dependency of the within-subject observations as well as a baseline correction between participants. Each dependent variable was included in a mixed model with fixed effect for the diet intervention and the race and stages (where applicable), including all two-way and three-way interactions. In the random effects structure of the model, we included a random intercept for the athletes to allow for differing levels of aerobic capacity when entering the training camp. Non-significant higher-order interactions were dropped from the model for ease of interpretation. The normality

assumption of the linear mixed model was assessed visually using QQ-plots of the model residuals. No obvious deviations of normality were detected. Tests for statistical significance of the fixed effects were performed using Type II Wald tests with Kenward-Roger degrees of freedom. 95% confidence intervals (CI) for the fixed effects were calculated using parametric bootstrap. Effect sizes based on the classical Cohen's d were calculated from the linear mixed model estimates, while accounting for the study design by using the square root of the sum of all the variance components (specified random effects and residual error) in the denominator.

Dietary intake data were analysed using SPSS Statistics 19 software (IBM, Armonk, New York, USA). Normality of data was checked with Shapiro-Wilk goodness-of-fit test and apart from CHO %E, total protein intake, and protein %E, data was normally distributed. For Adapt and De-Adapt phases involving comparison of three groups and two phases (intervention and adaptation), two-way ANOVA (group x phase) was used for normally distributed data with Tukey's post hoc tests, while Friedman's two way analysis of variance was used for data that were not normally distributed. Analyses of dietary intake associated with the 20 km race, involving comparison of two groups, were completed using a Student's t-test.

## Results

### Diet and training compliance–Adapt and de-adapt phases

Participants were compliant with their assigned dietary treatment and the monitoring of their food intake and training sessions. Table 3 summarises actual dietary intake and training volumes for the group who completed both the 25 d Adapt and the 10 d De-adapt phases of the study (n = 26), with the summary of mean daily intakes allowing the display of overall differences between diets over these periods. As intended, energy ($kJ \cdot kg^{-1} \cdot d^{-1}$) and protein ($g \cdot kg^{-1} \cdot d^{-1}$ and % energy) intake did not differ between dietary treatments over either phase. However, during the Adapt phase, daily fat ($g \cdot kg^{-1} \cdot d^{-1}$ and % of energy) and CHO ($g \cdot kg^{-1} \cdot d^{-1}$ and % energy) intakes were significantly skewed between the CHO-rich diets (HCHO and PCHO) and the LCHF diet. Although there was a similar mean daily intake of CHO with the HCHO and PCHO diets, intake was spread differently between and within some of the days, due to the timing of intake around training sessions designated as high and low CHO availability. The reported intakes from the self-chosen diets consumed during the taper/race preparation week are summarised in Table 4, along with the measured intake of fluid and CHO during the 20 km race.

Participants completed the 5.5-wk training intervention, with similar weekly training volumes (Table 3), although LCHF was associated with greater perception of effort and, on occasion, an inability to complete sessions as planned. This was consistently noted in daily wellbeing logs over the first 2 wk of the intervention but had returned to normal by Wk 3 [44]. Scale determined changes in BM over the Adapt phase showed a small but significant decrease in the LCHF group compared with the other groups (0.9 ± 0.9, 1.2 ± 1.0; 2.7 ± 1.5 kg, p = 0.008 for HCHO, PCHO and LCHF respectively). This difference disappeared with intake of CHO in De-Adapt, commensurate with restoration of muscle glycogen and water content. DXA-determined changes in body composition were small and similar between groups (data reported by Bone *et al.*, in review), suggesting that energy availability was preserved and standardised such that it did not interfere with training adaptations or favour one group over another.

### VO₂peak and economy testing

All race walkers participated in the graded economy and $VO_2$peak protocols at Baseline and Adapt testing, providing sample sizes of 8, 8 and 10 for HCHO, PCHO and LCHF,

**Table 3. Actual intake and training during 5 w intervention in elite race walkers involving 25 d adaptation to high carbohydrate (CHO) availability (HCHO), periodised CHO availability (PCHO) or low CHO high fat (LCHF) followed by 10 d period of HCHO de-adaptation.**

| Group | Nutrient | Unit | Adapt (25 d) | De-adapt (10 d) |
|---|---|---|---|---|
| HCHO (n = 8) | Energy | MJ·d$^{-1}$ | 14.0 ± 2.3 | 14.2 ± 2.1 |
| | | kJ·kg$^{-1}$·d$^{-1}$ | 223 ± 15 | 227 ± 20 |
| | Protein | g·d$^{-1}$ | 127 ± 23 | 139 ± 26 |
| | | g·kg$^{-1}$·d$^{-1}$ | 2.0 ± 0.2 | 2.2 ± 0.2 |
| | | %E | 15.0 ± 0.3 | 16 ± 1 |
| | CHO | g·d$^{-1}$ | 534 ± 77$^{*}$ | 503 ± 63 |
| | | g·kg$^{-1}$·d$^{-1}$ | 8.5 ± 0.4$^{*}$ | 8.1 ± 0.7 |
| | | %E | 65 ± 2$^{*\$}$ | 60 ± 1 |
| | Fat | g·d$^{-1}$ | 69 ± 16$^{*\$\$}$ | 83 ± 14 |
| | | g·kg$^{-1}$·d$^{-1}$ | 1.1 ± 0.1$^{*\$\$}$ | 1.3 ± 0.1 |
| | | %E | 18 ± 1$^{*\$\$\$}$ | 22 ± 1 |
| | Training | km (total) | 378 ± 67 | 126 ± 21 |
| PCHO (n = 8) | Energy | MJ·d$^{-1}$ | 13.3 ± 2.6$^{\$}$ | 13.9 ± 3.3 |
| | | kJ·kg$^{-1}$·d$^{-1}$ | 212 ± 21$^{\#\$}$ | 228 ± 33 |
| | Protein | g·d$^{-1}$ | 125 ± 24 | 134 ± 33 |
| | | g·kg$^{-1}$·d$^{-1}$ | 2.0 ± 0.2 | 2.2 ± 0.3 |
| | | %E | 16.0 ± 0.5$^{\#}$ | 16.0 ± 0.5 |
| | CHO | g·d$^{-1}$ | 490 ± 92$^{*}$ | 490 ± 112 |
| | | g·kg$^{-1}$·d$^{-1}$ | 7.8 ± 0.7$^{*\#}$ | 8.1 ± 1.1 |
| | | %E | 63 ± 1$^{\$}$ | 60 ± 1 |
| | Fat | g·d$^{-1}$ | 70 ± 17$^{*\$\$}$ | 82 ± 22 |
| | | g·kg$^{-1}$·d$^{-1}$ | 1.1 ± 0.2$^{*\$\$}$ | 1.3 ± 0.2 |
| | | %E | 20 ± 1$^{*\$\$\$}$ | 22 ± 1 |
| | Training | km (total) | 402 ± 85 | 157 ± 38 |
| LCHF (n = 10) | Energy | MJ·d$^{-1}$ | 15.4 ± 1.6 | 15.7 ± 1.7 |
| | | kJ·kg$^{-1}$·d$^{-1}$ | 234 ± 17 | 239 ± 27 |
| | Protein | g·d$^{-1}$ | 144 ± 18 | 151 ± 18 |
| | | g·kg$^{-1}$·d$^{-1}$ | 2.2 ± 0.2 | 2.3 ± 0.2 |
| | | %E | 16.0 ± 0.5$^{\#}$ | 16.0 ± 0.5 |
| | CHO | g·d$^{-1}$ | 35 ± 3$^{\$\$\$}$ | 552 ± 62 |
| | | g·kg$^{-1}$·d$^{-1}$ | 0.50 ± 0.05$^{\$\$\$}$ | 8.4 ± 1.0 |
| | | %E | 4.0 ± 0.2$^{¥\$\$}$ | 60 ± 1 |
| | Fat | g·d$^{-1}$ | 326 ± 34$^{\$\$\$}$ | 95 ± 12 |
| | | g·kg$^{-1}$·d$^{-1}$ | 4.9 ± 0.5$^{\$\$\$}$ | 1.4 ± 0.5 |
| | | %E | 78 ± 0.5$^{\$\$\$}$ | 22 ± 1 |
| | Training | km (total) | 424 ± 63 | 162 ± 21 |

$^{*}$ different from LCHF ($p < 0.001$)

$^{\#}$ different from HCHO ($p < 0.05$)

$^{¥}$ different from PCHO ($p < 0.05$).

$^{\$}$, $^{\$\$}$, $^{\$\$\$}$ different between Adapt and De-Adapt ($p < 0.05$, $p < 0.01$, $p < 0.001$).

respectively. The results of these tests are summarised in Table 5 (BM, HR, RER, RPE and VO$_2$peak data), Fig 3 (O$_2$ and substrate utilisation for the 2$^{nd}$ and 4$^{th}$ stage), and Fig 4 (blood metabolites). Changes in VO$_2$peak (L·min$^{-1}$) over the Adapt phase in each group were non-significant (p = 0.987). However, when combined with the small changes in BM and expressed in mL·kg$^{-1}$·min$^{-1}$, all groups achieved a significant increase over the Adapt phase: HCHO:

**Table 4. Self-chosen intake during race week by elite race walkers after 25 d high/periodised carbohydrate (CHO) availability (HCHO/PCHO) or low CHO high fat (LCHF) diet and 10 d HCHO de-adaptation.**

| Group | | | 6 d Taper | Pre-race meal | Race intake |
|---|---|---|---|---|---|
| PCHO/ HCHO+ HCHO (n = 11) | Energy | MJ·d$^{-1}$ | 12.3 ± 2.4 | 2.7 ± 1.2 | CHO 46 ± 21 g Fluid 337 ± 239 ml |
| | | kJ·kg$^{-1}$·d$^{-1}$ | 193 ± 45 | 42 ± 16 | |
| | Protein | g·d$^{-1}$ | 129 ± 25 | 16 ± 8 | |
| | | g·kg$^{-1}$·d$^{-1}$ | 2.0 ± 0.4 | | |
| | | %E | 18 ± 3 | | |
| | CHO | g·d$^{-1}$ | 368 ± 96 | 130 ± 59 | |
| | | g·kg$^{-1}$·d$^{-1}$ | 5.8 ± 1.6 | 2.0 ± 0.8 | |
| | | %E | 50 ± 4 | | |
| | Fat | g·d$^{-1}$ | 96 ± 21 | 8 ± 6 | |
| | | g·kg$^{-1}$·d$^{-1}$ | 1.5 ± 0.5 | | |
| | | %E | 29 ± 4 | | |
| | Training | km (total) | 42 ± 20 | | |
| LCHF +HCHO (n = 8) | Energy | MJ·d$^{-1}$ | 13.4 ± 2.8 | 2.7 ± 1.0 | CHO 52 ± 27 g Fluid 485 ± 239 ml |
| | | kJ·kg$^{-1}$·d$^{-1}$ | 207 ± 51 | 41 ± 12 | |
| | Protein | g·d$^{-1}$ | 153 ± 27 | 18 ± 7 | |
| | | g·kg$^{-1}$·d$^{-1}$ | 2.4 ± 0.5 | | |
| | | %E | 20 ± 4 | | |
| | CHO | g·d$^{-1}$ | 382 ± 103 | 110 ± 41 | |
| | | g·kg$^{-1}$·d$^{-1}$ | 5.9 ± 1.8 | 1.7 ± 0.5 | |
| | | %E | 48 ± 5 | | |
| | Fat | g·d$^{-1}$ | 109 ± 33 | 14 ± 11 | |
| | | g·kg$^{-1}$·d$^{-1}$ | 1.7 ± 0.5 | | |
| | | %E | 30 ± 5 | | |
| | Training | km (total) | 56 ± 13 | | |

57.6 ± 4.6 to 58.3 ± 4.1; PCHO: 58.5 ± 3.2 to 59.9 ± 3.7; LCHF: 61.3 ± 5.1 to 63.6 ± 4.0; $d = 0.36$, p = 0.03). The peak aerobic capacity of the LCHF group was higher than the other two groups at both Baseline and Adapt testing (p = 0.05).

At both testing times, the increase in exercise intensities across the stages of the economy test was associated with an increase in RER (p<0.001), VO$_2$peak (p<0.001), HR (p<0.001) and RPE (p<0.001) [Table 5]. There were significant diet vs test interactions for RPE and HR, with the LCHF group displaying higher values in the Adapt trial (p< 0.001), indicating a higher metabolic cost and perceived effort with this treatment compared with baseline testing and the other diets (Table 5). HR values for the HCHO and PCHO were decreased in the Adapt testing (p<0.001). There was a significant decrease in post-treatment RER values in the LCHF group (p<0.001) compared with the pre-treatment trial across the four economy stages. Differences in pre- to post-treatment values for HCHO (p>0.999) and PCHO groups (p = 0.184) were minor. The absolute O$_2$ cost of exercise (L·min$^{-1}$) increased across the four economy stages in all groups both at Baseline and Adapt testing [Table 5]. However, at Adapt testing, values for the LCHF group were higher than for Baseline (p< 0.001), while they were maintained at similar levels for HCHO and PCHO groups. Separate analysis of characteristics at the completion of the VO$_2$peak test revealed a lower RER in Adapt testing compared with Baseline values in the LCHF group ($d = 3.29$, p = 0.04). In addition, HR at this same point in the Adapt trial was lower than in the Baseline trial for the HCHO ($d = 0.51$) and PCHO ($d = 0.46$) groups (p<0.001).

**Table 5. Four stage graded economy and maximal aerobic capacity test before (Baseline) and after (Adapt) 25 d adaptation to high carbohydrate (HCHO)/periodised CHO (PCHO) availability or low CHO high fat (LCHF) diet in elite race walkers (n = 26).**

| | | HCHO (n = 8) | | | | | PCHO (n = 8) | | | | | LCHF (n = 10) | | | | |
|---|---|---|---|---|---|---|---|---|---|---|---|---|---|---|---|---|
| | | S1 | S2 | S3 | S4 | Peak | S1 | S2 | S3 | S4 | Max | S1 | S2 | S3 | S4 | Peak |
| **Body mass (kg)** | **Baseline** | 63.3 ±9.0 | | | | | 63.0 ±10.0 | | | | | 66.6 ±7.6 | | | | |
| | **Adapt** | 62.4 ±8.4 | | | | | 61.9 ±9.6 | | | | | 63.8 ±6.9[*] | | | | |
| Respiratory Exchange Ratio | Baseline | 0.87 ±0.05 | 0.91 ±0.05 | 0.96 ±0.04 | 0.99 ±0.03 | 1.06 ±0.04 | 0.82 ±0.05 | 0.87 ±0.05 | 0.92 ±0.04 | 0.96 ±0.04 | 1.08 ±0.06 | 0.85 ±0.04 | 0.89 ±0.04 | 0.93 ±0.04 | 0.97 ±0.03 | 1.09 ±0.03 |
| | Adapt[#] | 0.86 ±0.02 | 0.91 ±0.02 | 0.96 ±0.04 | 0.98 ±0.04 | 1.09 ±0.05 | 0.84 ±0.04 | 0.89 ±0.04 | 0.93 ±0.03 | 0.98 ±0.02 | 1.11 ±0.04 | 0.73 ±0.03 | 0.77 ±0.02 | 0.81 ±0.02 | 0.86 ±0.04 | 0.94[*] ±0.03 |
| $VO_2$ (L·min⁻¹) | Baseline | 2.50 ±0.58 | 2.82 ±0.61 | 3.11 ±0.66 | 3.38 ±0.66 | 3.66 ±0.65 | 2.47 ±0.45 | 2.84 ±0.48 | 3.13 ±0.52 | 3.38 ±0.56 | 3.70 ±0.67 | 2.74 ±0.43 | 3.11 ±0.45 | 3.44 ±0.49 | 3.73 ±0.54 | 4.09 ±0.61 |
| | Adapt[@] | 2.41 ±0.56 | 2.74 ±0.61 | 3.05 ±0.65 | 3.30 ±0.67 | 3.64 ±0.55 | 2.30 ±0.39 | 2.71 ±0.48 | 2.94 ±0.46 | 3.25 ±0.51 | 3.69 ±0.53 | 2.82 ±0.41 | 3.20 ±0.44 | 3.54 ±0.49 | 3.80 ±0.49 | 4.07 ±0.53 |
| Heart Rate (b·min⁻¹) | Baseline | 145±8 | 157±8 | 170±9 | 178 ±10 | 185 ±12 | 144 ±10 | 159±9 | 170 ±10 | 177 ±11 | 186±8 | 143 ±10 | 156±8 | 167±8 | 173±8 | 185±8 |
| | Adapt[@&] | 133±9 | 146±9 | 158±8 | 167±8 | 180[*] ±11 | 135 ±10 | 148 ±11 | 158 ±12 | 167 ±11 | 182[*] ±9 | 149±8 | 160±7 | 169±6 | 175±7 | 184±8 |
| RPE | Baseline | 9.8 ±1.8 | 11.6 ±1.7 | 13.4 ±1.6 | 14.8 ±1.6 | | 8.6 ±1.9 | 10.4 ±1.9 | 12.5 ±2.2 | 14.9 ±2.0 | | 9.4 ±1.7 | 10.6 ±1.7 | 11.8 ±1.9 | 14.0 ±1.8 | |
| | Adapt[#] | 10.0 ±1.6 | 12.1 ±1.9 | 13.8 ±2.1 | 15.3 ±2.1 | | 8.9 ±1.6 | 10.6 ±1.6 | 12.9 ±1.6 | 14.5 ±1.8 | | 11.0 ±1.2 | 12.4 ±0.8 | 14.2 ±1.4 | 16.6 ±1.6 | |

Data are mean (SD) and statistical comparisons note differences across the intervention (Baseline to Adapt) and between groups receiving high carbohydrate availability (HCHO), periodised CHO availability (PCHO) and low carbohydrate high fat (LCHF) dietary treatments.

[*] Difference between Baseline and Adapt values within the group

@Higher in Adapt trial across stages 1–4 than Baseline trial for LCHF (p < 0.001)

#Lower in Adapt trial across stages 1–4 than in Baseline trial for LCHF (p <0.001).

&Lower in Adapt trial across stages 1–4 than Baseline trial for HCHO and PCHO (p < 0.001).

Fig 3 illustrates the changes in substrate utilisation and the $O_2$ cost of exercise due to diet and/or training by focusing on stage 2 (Fig 3A) and Stage 4 (Fig 3B) of the economy test; these stages correspond to the typical walking speeds of elite race walkers during a 50-km event and 20-km event respectively. At the second stage, there was a significant increase in the relative $O_2$ cost of exercise (mL·kg⁻¹·min⁻¹) in the LCHF group in the Adapt trial ($d$ = 0.81, p = 0.036), while this remained similar in the PCHO (p = 0.999) and HCHO group (p < 0.999). The pattern was repeated in the results from the fourth stage of the economy test: there was greater increase in the relative $O_2$ cost of exercise (mL·kg⁻¹·min⁻¹) in the LCHF group in the Adapt trial ($d$ = 0.84, p = 0.021), while this remained constant in the PCHO (p< 0.999) and HCHO group (p< 0.999). The substrate utilisation data show an increase in rates of CHO oxidation and a reduction in rates of fat oxidation across all stages in all groups for both Baseline and Adapt testing protocols (p < 0.001). Rates of CHO oxidation were decreased and fat oxidation was increased in Adapt testing for the LCHF group (p<0.001), but remained similar for the HCHO and PCHO groups between tests.

Fig 4 summarises concentrations of metabolites (glucose, lactate and β-hydroxybutyrate) in capillary blood samples obtained during the economy test, representing concentrations at the end of each stage and at the conclusion of the maximal aerobic capacity component. Blood glucose concentrations increased from baseline values during exercise, with an increase in exercise intensity being associated with high blood glucose concentrations (Stages 1, 2, 3 < 4, Max; p<0.001). Across HCHO and PCHO groups, there was a decrease in blood glucose concentrations across the training intervention such that values in the Adapt trials were lower than in

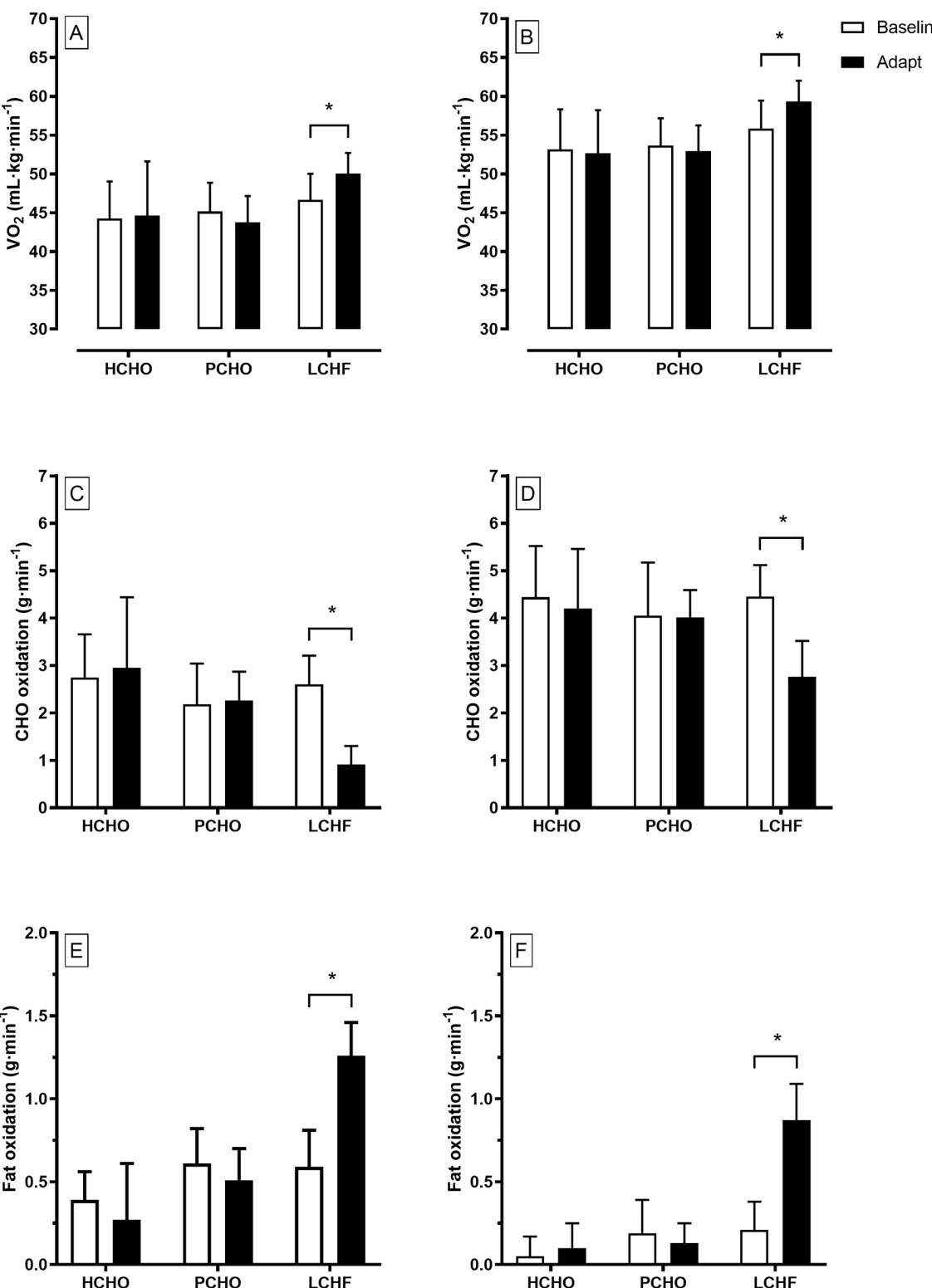

**Fig 3. Oxygen uptake (mL·kg$^{-1}$·min$^{-1}$), and rates of CHO oxidation (g·min$^{-1}$) and fat oxidation (g·min$^{-1}$) during graded economy test at 2nd stage approximating 50 km race speed (A) and 4th stage (B) approximating 20 km race speed in elite race walkers pre- (Baseline) and post (Adapt)- 25 d of intensified training and high carbohydrate availability (HCHO, n = 8); periodised carbohydrate availability (PCHO, n = 8) or ketogenic low carbohydrate high fat (LCHF, n = 10) diets.** B: * = significantly different pre-treatment (p< 0.01).

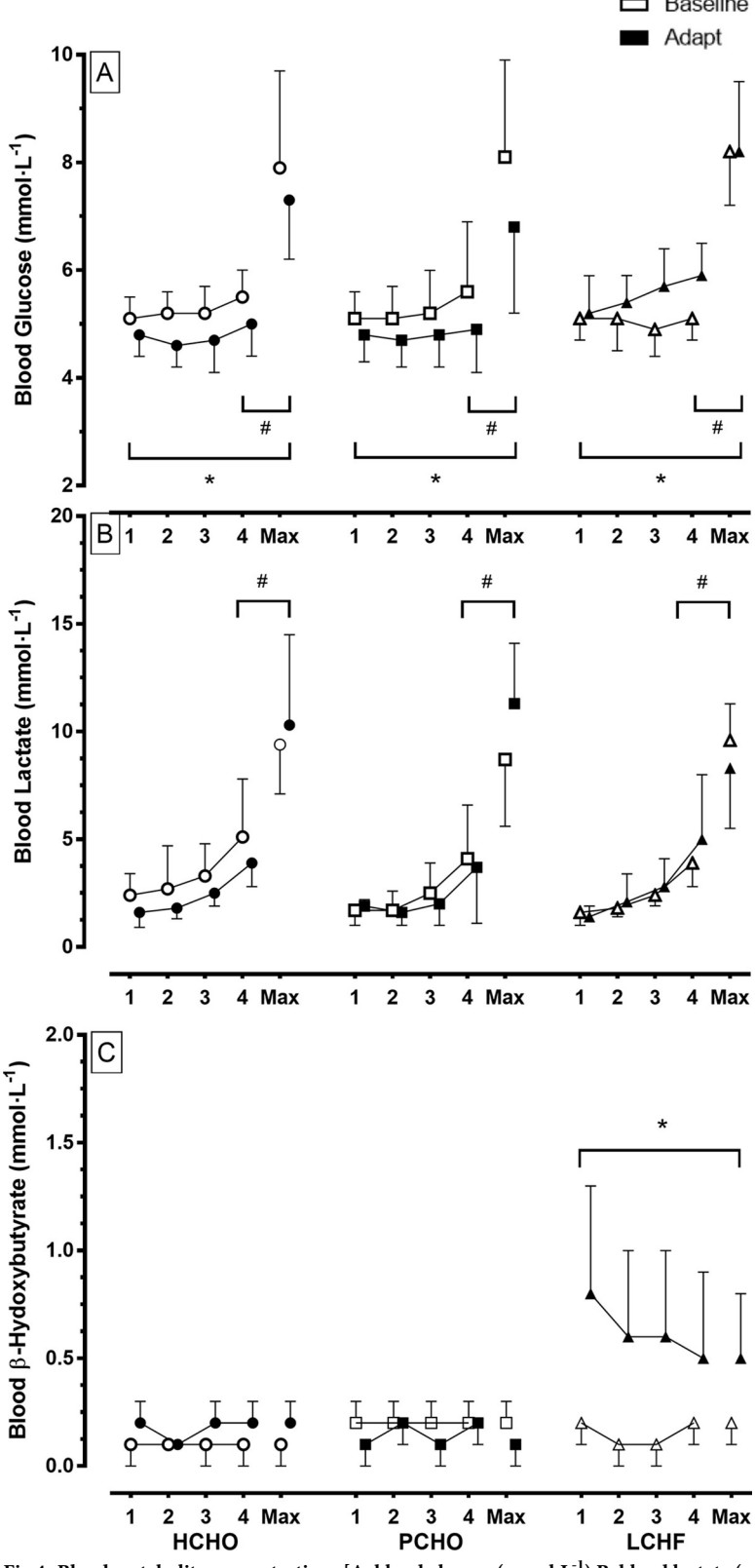

**Fig 4. Blood metabolite concentrations [A: blood glucose (mmol·L$^{-1}$) B: blood lactate (mmol·L$^{-1}$) and C: blood ß-hydroxybutyrate (mmol·L$^{-1}$)] during graded economy test and test for peak aerobic capacity in elite race walkers pre- and post- 25 d of intensified training and high carbohydrate availability (HCHO, n = 8); periodised**

carbohydrate availability (PCHO, n = 8) or ketogenic low carbohydrate high fat (LCHF, n = 10) diets. * = significantly different to pre-treatment (p< 0.01).

the Baseline trial (p<0.001); however concentrations in the LCHF trial were increased (p<0.001). Blood lactate concentrations increased as exercise intensity was increased across all dietary treatments (Stages 1, 2, 3 < 4, Max; p<0.001), with similar patterns in both trials. Blood β-hydroxybutyrate was maintained at low concentrations across the economy test for both Baseline and Adapt trials with the HCHO and PCHO groups. However, the Adapt trial in the LCHF group was associated with a significant increase in blood β-hydroxybutyrate concentrations compared with the Baseline trial (p<0.001).

## 10,000 m race

The study produced 25 data sets in which a race walker competed in a 10,000 m race pre- and post- their allocated dietary intervention. One participant from the PCHO group experienced an injury which prevented him from competing in the second race. Race performance data therefore represent n = 8, 7 and 10 for HCHO, PCHO and LCHF, respectively. Environmental conditions at the commencement of Race 1 (Baseline) were 24°C, 58% relative humidity (r.h.) and wind speed: 1 m·s$^{-1}$, with those of Race 2 (Adapt) being: 27.5°C, 49% r.h., 0.6 m·s$^{-1}$. Finishing times for the 10,000 m races are summarised in Fig 5A. The HCHO and PCHO groups completed Race 2 in times that were significantly and marginally faster than Race 1, respectively, with a 4.8% improvement (equivalent to 134 s, 95% CI: [62 to 207 s]; $d$ = 0.84, p <0.001) and 2.2% (61 s, [-18 to 144 s]; $d$ = 0.38, p = 0.09] improvement in performance following the 3-wk diet and training intervention, respectively. Differences in completion times between Race 1 and Race 2 in the LCHF group showed a significant performance change (impairment) of—3.3% (86 s: [-18 to -144 s]; $d$ = 0.54, p < 0.001)

Capillary blood samples collected pre-race and post-race showed a similar increase in blood glucose after the race (~10 mmol·L$^{-1}$) compared with pre-race values (~5–6 mmol·L$^{-1}$) in all groups and races Blood lactate concentrations showed a similar pattern across groups for all races, with pre-race and post-race showed a similar increase in blood glucose after the race (~8–10 mmol·L$^{-1}$) compared with pre-race values (~1.5 mmol·L$^{-1}$). Blood β-hydroxybutyrate concentrations were low at pre- (~0.1 mmol·L$^{-1}$) and post-race (~0.3 mmol·L$^{-1}$) time points in all groups, apart from Race 2 in the LCHF group where they were higher than in the other groups at pre- (0.3 mmol·L$^{-1}$) and post-race (0.6 mmol·L$^{-1}$).

## 20 km race performance

Performance during the 20 km Australian national Race Walking championships for the 19 walkers who completed the race is summarised in Fig 5B and 5C. Environmental conditions for the race were 12–15°C, continual light rain with ~70% r.h. and wind speed: 0.5–1.5 m·s$^{-1}$. Overall, the group completed the race in times that represented 100.2 ± 1.5% of their previous personal bests; 7 participants achieved a new personal best time at the race (4 from CHO/ HCHO and 3 from LCHF/HCHO) groups. Comparison of performances in 10,000 m race walks in this cohort showed that they mirrored the outcomes of the larger group with the CHO/HCHO subgroup (n = 11) showing a 2.3% improvement in Race 2 (Adapt) vs Race 1 (Baseline) while the LCHF /HCHO subgroup recorded a performance change (impairment) of 3.5%

The 20 km performances of these groups represented 191 ± 6% and 197 ± 6% of the times of the Race 1 time for CHO/HCHO and LCHF/HCHO respectively. Comparison of these data

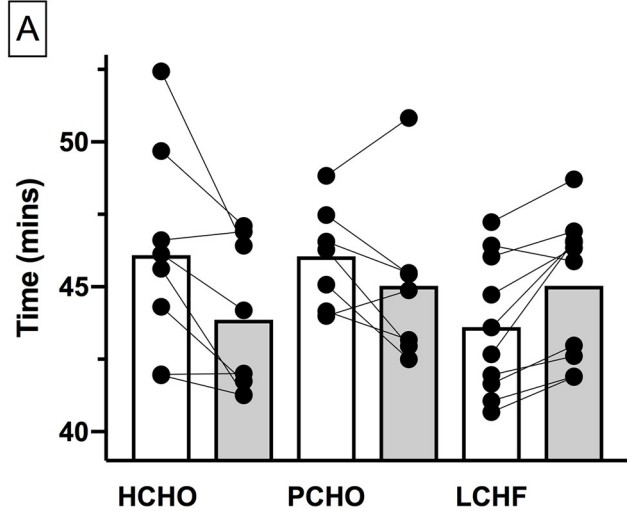

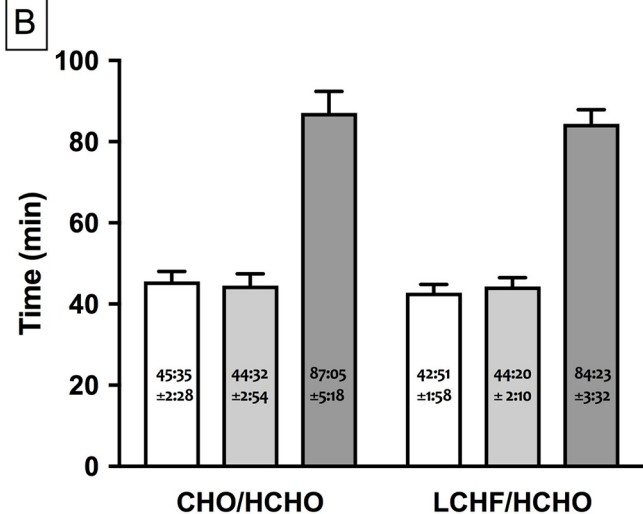

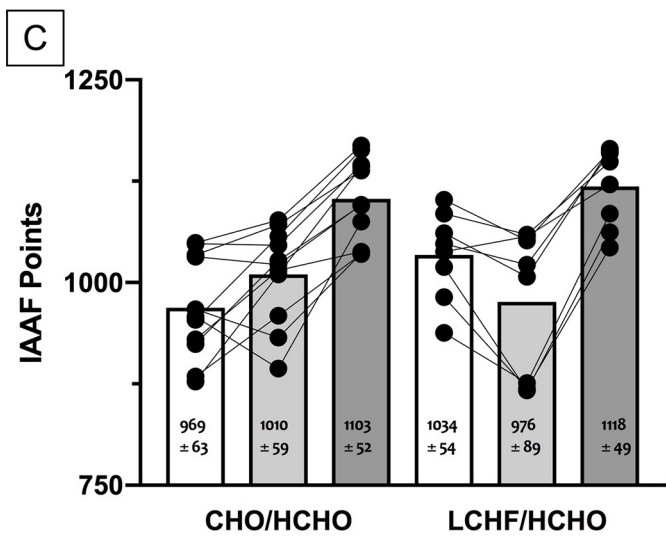

**Fig 5. Race times for IAAF sanctioned events completed during study.** A: 10,000 m race walk events in elite race walkers undertaken pre- (Race 1; Baseline) and post- (Race 2; Adapt) 25 d of intensified training and high carbohydrate availability (HCHO, n = 8); periodised carbohydrate availability (PCHO, n = 7) or ketogenic low carbohydrate high fat (LCHF, n = 10) diets. B: comparison of 10,000 m and 20 km race times for subgroup of participants who undertook 3 wk of de-adapt to HCHO diet and race taper after 25 d intensified training and HCHO or PCHO diet (n = 11) or LCHF diet (n = 8) and C: comparison of 10,000 and 20 km race outcomes of this subgroup expressed as IAAF ranking points. * = significantly faster than Race 1 (p< 0.01); # = significantly slower than Race 1 (p< 0.01).

for the three races showed that CHO/HCHO improved from Race 1 to Race 2 with no further improvement to the 20 km race, whereas the LCHF/HCHO recorded a significantly slower performance in Race 2, and their 20 km race was slower than would be predicted from Race 1 (p<0.01) [Fig 5B]. Comparison of these race performances based on IAAF Points (Fig 5C) showed that the CHO/HCHO group performed better in Race 2 compared with Race 1 and maintained a similar performance level for the 20 km race. Meanwhile the LCHF/HCHO group commenced with a significantly higher performance for Race 1 than the other group (p = 0.011), showed a decrease in performance in Race 2 (p<0.001), but improved by the 20 km race such that their overall gain from Race 1 was similar to that achieved by the CHO/HCHO group.

## Discussion

This study directly replicated and extended an earlier investigation of the effect of a ketogenic low-CHO, high-fat diet, consumed during a period of intensified training by elite athletes, on exercise characteristics across the range of intensities at which endurance and ultra-endurance athletes train and compete [28, 45–47]. The protocol repeated major elements of the previous work, with alterations that were systematically identified from lay and peer responses to the original investigation to enhance the methodology or address unanswered issues. The novel findings were: (1) Key outcomes of the earlier study were reproduced in another cohort of elite race walkers that included female participants: rates of fat oxidation during exercise of varying speeds/intensities were substantially increased by exposure to the LCHF diet, with an associated increase in the oxygen cost of such exercise (i.e., reduced economy) as predicted from the stochiometry of fat and CHO oxidation pathways, (2) Training and racing while consuming diets providing periodic or sustained high CHO availability improved real-world endurance performance in contrast to the performance decrement on a LCHF diet, and (3). There was no evidence of purported performance benefits associated with a specific model of periodised exposure to LCHF (i.e., adaptation to LCHF followed by return to a diet with high CHO availability during the final preparation and taper for competition).

There is a call for the replication of studies in biomedical fields, particularly those involving small sample sizes, to investigate the reproducibility of findings [3, 48–50]. We consider our study to approach an independent direct replication design, due to the significant number of new personnel within the research team, and the involvement of new (19/26) participants including females within the cohort of elite race walkers, or allocation to new treatments (3/7, with 2, 1 and 1 participants electing to repeat their previous intervention with LCHF, PCHO and HCHO diets, respectively). The design involved substantial repetition of protocols undertaken during the earlier study with several enhancements due to scrutiny of the previous protocol (e.g., completion as a single research camp to remove or reduce the potential influence of effects such as different environmental conditions for races or inter-athlete dynamics during training sessions). Protocol extensions investigated a hypothesis generated by others from anecdotal observations around the earlier study; that superior long term performance

outcomes would be achieved by adapting to a LCHF diet before returning to a CHO-rich diet and race taper than by chronic dietary support based on high CHO availability. Unlike the majority of replication studies across different fields in which findings are either not reproduced, or are reproduced with a reduction in the magnitude of the effects [44–46], the main outcomes of our earlier study were confirmed. Since some of these variables (e.g., race performance by elite athletes in sanctioned events) can be affected by a number of other influences outside even the best study control or standardization, the consistency of our findings in small sample sizes provides us with confidence around the robustness of the effects.

The inclusion of female participants was a deliberate element of the new study; this adds to the ecological validity of the study but created a risk of diluted findings due to the greater numerical spread of the data (e.g., female athletes have lower values for aerobic capacity and speed), as well as the potential for females to have a different response due to true sex differences or a confounding effect of menstrual status on important variables. The study design (a single training camp) prevented us from controlling or standardizing the phase of the menstrual cycle other than noting that the protocol created a period of ~3.5 wk between the Baseline and Adapt testing, and therefore the likelihood of it falling during a similar menstrual phase within each female participant. The low participant numbers, the non-randomised allocation of the dietary treatments, the confounding "camp effect" of chronically training with partners of superior ability, and the loss of one of the two females from the LCHF treatment precludes any meaningful discussion of sex-specific responses to the treatment. Indeed, the only warranted statement is that the response to the LCHF treatment is sufficiently robust that it overrides any variability in the data due to the inclusion of females *per se* or several associated potentially confounding variables.

Our two studies show remarkable consistency in the outcomes of adaptation to a LCHF diet on the real world performance of the 10,000 m race walking event; a race characterized as sustained high-intensity endurance exercise. One advantage of the new protocol involving a single training camp was that all athletes competed in the same two races under the same environmental and external factors, and with equal psychological investment in their preceding treatment. The results showed a mean gap of ~7% in the performance differential from Race 1 to Race 2 between the LCHF athletes and those who trained and raced with CHO support, with the differences reaching statistical significance for the HCHO group (race improvement by nearly 5%) and LCHF cohort (race impairment by ~2%). A similar mean margin of ~8% was reported in our earlier study, although race times were also affected by different weather conditions over 4 races such that the differences equated to a significant 6% improvement with the HCHO treatment and a negligible impairment of -1.6% with the LCHF treatment. These performance differences are robust and highly meaningful to the outcomes of real life sporting events.

In explaining these changes in performance, it is necessary to fully appreciate the changes in fuel utilisation during exercise achieved by the LCHF diet and our theory that the benefits claimed by the proponents of this diet are, in fact, the cause of the impaired capacity for higher-intensity aerobic exercise. A key rationale for the uptake of the LCHF diet by endurance athletes is a substantial increase in the capacity to oxidise fat as a muscle substrate across an increased range of exercise intensities [5, 6, 26, 27]. We found that 25 d of rigorously controlled intake of the ketogenic (~0.5 $g \cdot kg^{-1} \cdot d$-1 CHO, 18% protein as energy, 75–80% fat) diet [17] was associated with a doubling of fat oxidation at relevant exercise intensities (e.g. an increase from $0.59 \pm 0.22$ to $1.26 \pm 0.20$ $g \cdot min^{-1}$ at a speed approximating 50 km race pace), in elite endurance athletes who already exhibit superior fat oxidation capacity than sedentary or lesser trained populations [51], due to their extensive training history, and, potentially, genetic predisposition. These findings concur with observations from our earlier study, where rates of fat oxidation increased from $0.58 \pm 0.22$ to $1.38 \pm 0.26$ $g \cdot min^{-1}$ [4], particularly when the

inclusion of a female participant of smaller size and slower absolute speed (i.e., power output) is taken into account. These absolute rates of fat oxidation are of the same magnitude as those reported in cross-sectional studies of athletes who have undertaken such dietary treatments for > 8 mo [25, 26]. Because we used an event-specific 4-stage graded exercise test that is widely employed in the testing of high performance athletes [52], rather than the cycling/ treadmill running fat max test with a larger number of separate work intensities [53, 54], we were unable to directly measure the exercise intensity at which maximal rates of fat oxidation ($Fat_{max}$) occur. However, it is likely that our intervention would have shifted $Fat_{max}$ from the typical ~50–65% of peak aerobic capacity to values of ~65–75% $VO_2max$, as demonstrated in populations with longer-term keto-adaptation [26].

The current study did not allow the interrogation of muscle substrate kinetics or mechanisms underpinning the changes in fuel use during exercise. However, we note the similarity of changes in blood metabolites and substrate utilization in the LCHF group across our studies, and refer readers to other reviews of the mechanisms underpinning changes in fat oxidation and carbohydrate metabolism following adaptation to high fat diets [25, 55]. As in the previous study, keto-adaptation was associated with a significant and sustained increase in circulating β-hydroxybutyrate bodies and, presumably, an increased contribution of these metabolites to energy expenditure at rest and during exercise. The methodology employed in our studies, and others which have examined metabolism during exercise in keto-adapted athletes [10, 26, 27] have not accounted for their contribution to substrate use. Although this could be seen as a systematic error that does not change the assessment of the oxygen cost of exercise, we acknowledge that in some populations, enhancements in training status might lead to an increase in muscle oxidation of ketone bodies leading to a greater underestimation of their contribution in the post-intervention phase [56].

Increased rates of fat oxidation during exercise are a hallmark of endurance training and offer the advantage of making better use of a fuel substrate found in relatively unlimited amounts in even the leanest endurance athlete [25]. Furthermore, we appreciate that a capacity for high rates of fat oxidation in fasted but non-adapted athletes in a $Fat_{max}$ test may correlate with race performance, as has been recently reported in ultra-endurance triathletes [54]. This finding is consistent with the model we have used to explain our study findings, if we consider that rates of fat oxidation might be a proxy for absolute work capacity and mitochondrial mass in one setting, but that competition outcomes are achieved with different nutritional strategies. Our model, based on findings from our earlier study and reproduced here, is that the economy (defined as the oxygen cost of a given speed or power output) of exercise undertaken with increased utilisation or reliance on fat oxidation is reduced by small, but measurable and crucial, amounts. Empirically derived data from over a century ago [57, 58] or more recently [59] have demonstrated the higher oxygen cost of deriving adenosine triphosphate (ATP) from fat rather than CHO fuels; this can be explained by an appreciation of the stoichiometry of oxidative phosphorylation from different substrates [60]. Indeed, another group who studied 31 d of LCHF in endurance-trained runners confirmed our findings of impaired exercise economy, but noted that this effect was amplified at intensities above 70% $VO_2max$ and was of greater magnitude than could be explained by stoichiometric calculations [61]. They speculated that mitochondrial uncoupling, as observed by our research team in a separate study of a non-ketogenic high-fat diet and skeletal muscle mitochondrial respiration [62], might also contribute to reduced ATP production per oxidative cost [61]. Here, we found, via an *in vitro* substrate-uncoupler-inhibition-titration technique in permeabilised muscle fibres, that 5 d of adaptation to a high fat diet reduced mitochondrial respiration supported by octanoylcarnitine and pyruvate as well as uncoupled respiration at rest, without change in the protein abundance of the complexes [62].

Yet another mechanism that might contribute to reduced exercise economy with the LCHF diet involves changes in the oral [63] and stool [64] microbiome diet reported from our first study [4]; these included a decrease in the relative abundance of gram Negative nitrate-nitrite reducing bacteria in the mouth [63], while stool samples showed a significant reduction in Faecalibacterium species and an increase in Bacteroides and Dorea species were increased [64]. Whether and how these observations are associated with health and performance remains to be determined; however, we reported a correlation between the relative abundance of some enterotype species and rates of fat oxidation or walking economy [64]. Furthermore, interruption to colonies of nitrate/nitrate reducing bacterial in the mouth due to the use of oral mouthwashes has been shown to disrupt the production of Nitric Oxide (NO) from dietary nitrate via the entero-salivary pathway, causing functional endpoints (e.g., increase in blood pressure) [65]. Noting that nitrate/beetroot juice supplements are popularly used by athletes as a performance aid due to their demonstrated ability to decrease the oxygen cost of exercise [66], it is possible that the LCHF diet might achieve the reverse by reducing the operation of this pathway and its ability to produce NO under conditions of hypoxia and acidosis as found during high-intensity exercise [66].

Whatever the underlying cause of the increased oxygen cost of exercise, we note that the main characteristics of successful endurance athletes are a high aerobic capacity, the ability to sustain exercise for long periods at a high percentage of this, and high economy of movement [44, 67]. Indeed, both modelling [68] and measures [69] of exercise economy in athletes show its importance in determining performance and discriminating between successful athletes [70]. According to our model, the increase in fat oxidation associated with keto-adaptation increases the oxygen cost of movement at any given speed and, we postulate, reduces the speed or power achieved at any given percentage of maximal aerobic capacity. Neither of our studies, nor those which have assessed longer-term adaptation to a ketogenic LCHF [26,27] have directly assessed this important hypothesis. Yet, it yields important considerations for training and competing in endurance sports; notwithstanding the sparing of glycogen associated with higher fat oxidation rates, athletes must exercise at a higher percentage of their aerobic capacity and at a higher heart rate to achieve the same speed or power outputs, or accept a lower output when working at the same fraction of their aerobic or cardiovascular capacity. The implications of this concept are supported by consistent findings of a preservation of the ability to perform exercise at lower workloads, where there is reserve to absorb the increase in aerobic cost following adaptation to a high fat diet [10, 61], but a reduction in capacity for exercise at higher intensities around or above the so-called anaerobic threshold [4, 10, 15, 61]. To put the observed change in exercise economy with the LCHF diet into context, in the current study, the oxygen cost (ml.kg$^{-1}$.min$^{-1}$) of walking at ~20 km race speed during the economy test was increased by 7.1%, while the 50 km race speed was associated with a 6.2% increase in oxygen cost. This compares with values of 7.5% and 6.2% for the changes seen in the first study [4]. By contrast, the world of distance running is currently enamoured and/or outraged by improvements in performance achieved by new shoe technology [71], which achieves a 4% mean reduction in the oxygen cost of running [72].

Of course, it is important to consider that other factors might also have contributed to our reproduced findings of sub-optimal race performance following keto-adaptation. Although interruption to training quality and perceptions of exertion were reported in the first 2 weeks of the LCHF treatment, we noted that training effort and gross metrics (e.g. overall volume and the completion of key sessions) were restored for the second half of the training camp. This time lag between upregulation of fat oxidation and the abatement of fatigue has been previously reported [10]. Furthermore, the LCHF group achieved a training enhancement in the form of an increase in aerobic capacity (ml·kg$^{-1}$·min$^{-1}$) and a greater reduction in BM

(presumably due to loss of muscle glycogen/water) than the other groups which should have reduced the energy cost of race walking. However, this provides further evidence that their failure to achieve a performance improvement is at least partially explained by the decreased speed hypothesised to occur at their highest sustainable fraction of aerobic capacity, due to the lower economy of fat oxidation and the known impairment of muscle glycogen oxidation [73].

The practical application of our hypothesis is that consideration of the value of adaptation to a LCHF diet may require an audit of whether an inability to maintain adequate CHO availability via glycogen supercompensation and rates of intake of exogenous CHO is of greater risk than the need to compete at intensities that are limited by the aerobic cost of high rates of fat oxidation, either throughout the event or at critical stages. Which sporting events meet such a categorisation is a topic of debate [46, 74], but should also consider the characteristics of the individual athlete and their responsiveness to and tolerance of the LCHF diet, since this is known to be variable [25, 27, 61]. Alternatively, strategies that try to periodise the exposure to high CHO availability against the background of superior capacity for fat oxidation have been proposed as a method to reduce the limitations of fat as an exercise substrate [18]. Our present study tested the anecdotal proposition that prior exposure to the LCHF diet acts like altitude training in leaving a legacy of physiological adaptations that would integrate with restoration of high CHO availability and a taper to produce superior race performance. Interrogation of performance in a 20 km race undertaken after such periodization (~ 3.5 wk LCHF diet followed by ~2.5 wk of return to HCHO) failed to find any evidence of performance enhancement; indeed, one method of analysis (normalising the outcome mathematically against the results of a 10,000 m race) suggested that the cohort who undertook this protocol achieved race results that were slower than expected from their pre-adaptation race results. It is worth noting that within our study conditions, 7 athletes achieved a personal best in the 20 km road race, with roughly equal numbers (4/11 and 3/9 from CHO/HCHO and LCHF/HCHO, respectively) achieving this outcome. It is expected, and a sign of the authenticity of the study design, that elite athletes who continue to refine their training program (such as exposing themselves to the "training camp" effect) and receive adequate motivation (prize money, qualification to international events, personal recognition etc) will be able to achieve incremental improvements in performance.

Many other permutations and combinations of periodization of fat-adaptation with episodes of high CHO availability are possible and merit investigation. For example, a recent case history [75] detailed the experiences of a highly competitive triathlete who incorporated 8 sessions in which CHO (60 g·h⁻¹) was consumed during high intensity training sessions (2·wk⁻¹) to his long-term (> 2 y) LCHF diet. Compared to his habitual diet, the introduction of exogenous CHO intake to performance trials, in conjunction with practice in training, was associated with meaningful (2.8% and 1.6%) improvements in a 20 km cycling time trial and a 4-min power test, respectively, while changes in 30 s sprint and 100 km time trial were judged to be trivial [70]. This experience supports the importance of high CHO availability for sustained higher-intensity efforts but requires further investigation of the best overall strategy for training and racing support. Similarly, in fairness to claims for the benefits of the LCHF diet, we acknowledge the limitations of the current study, particularly the inevitable criticism that its duration was insufficient to elicit the full benefits of longer-term keto-adaptation. However, as detailed in Table 1, our protocol exceeded the period shown by several different laboratories [12–15] to achieve robust cellular retooling to maximise rates of muscle fat oxidation, as well as achieve training-associated performance changes [30]. We also removed the separate effects of significant changes in body composition often seen in studies of self-managed implementation of LCHF diets [24]. Nevertheless, we cannot rule out the possibility that long-term exposure to high levels of circulating ketone bodies produces effects other than those associated

with adaptation to high fat intake/CHO restriction. For example, supplementation with exogenous ketone products to achieve similarly high levels has been shown to alter metabolism and enhance performance in one study [76], although this has been a singular fining [77–80]. Neither can we be sure that chronic CHO restriction will not exacerbate the impairments to the interaction of exercise with body systems such as the immune system [81], bone remodeling [82] and iron metabolism [83] seen with acute episodes of low CHO availability, or interfere with long term training quality by reducing the capacity to undertake higher-intensity workouts. Therefore, longer-term studies are welcomed but require careful control so that extraneous factors are removed.

The final element of this replication study involved the investigation of periodised CHO availability in which a series of different strategies of high and low CHO availability were integrated and sequenced into the training program to amplify the various desired outcomes of key sessions [18]. Strategies to achieve high CHO availability were clustered to support workouts of sustained or repeated intervals of higher intensity walking, or to develop gut absorption and tolerance during prolonged session simulating race conditions [18]. Meanwhile, delayed restoration of glycogen was organised to prolong the period of upregulated cellular signaling in response to some key exercise sessions [33, 35, 36] as well as to manipulate low glycogen concentrations and exogenous CHO for the subsequent session in which an amplification of the exercise stress could be expected [34, 35]. Again, we were able to reproduce the findings of our earlier study in failing to see a superior performance outcome from this periodised approach to CHO availability above that of the diet with chronic high CHO availability; indeed, a lower magnitude of improvement was seen in both studies in the PCHO group. These findings contradict the outcomes of previous work in well-trained but sub-elite triathletes which reported that a 3-wk intervention involving sequenced periodization of CHO availability produced performance improvements not seen when training was supported by the more traditional diet of sustained high CHO availability [84]. However, there is independent corroboration of our observations from a training study involving elite cyclists and triathletes, over a similar time frame and mode of periodised CHO availability, in which this treatment failed to provide superior improvements in muscle adaptations and performance than observed in a cohort receiving chronic high CHO support [85].

The lack of clear evidence for benefits of this "smarter" approach to training support in elite athletes is curious. Possible explanations include the implementation of our intervention in the base phase of the training program (where the increased training stimulus may have already maximised the adaptive response and reduced the capacity for differences due to CHO availability) or the large training volumes (which might have depleted CHO stores even in the face of the high CHO intakes around training such the actual differential between the HCHO and PCHO interventions was reduced). It is also possible that the additional fatigue associated with "train low" sessions might have masked superior adaptations, and like our tested theory around the time lag of the benefit of the LCHF, might need a period of training taper to abate the fatigue and allow true performance benefits to be revealed. The low sample size of the cohort in the present study who continued on to race in the 20-km championships prevents us from making a meaningful separate analysis of the subgroup who had undertaken PCHO treatment in the Adapt phase of our intervention versus HCHO. However, pooling the results from the current study with observations of the real life outcomes following our 2016 study of race walkers who self-selected to compete in the same race, but after the completion of that study, fails to find any evidence of superior performances from the PCHO subgroup. Further work is needed to address the anomaly around the benefits of the periodization of CHO availability in elite athletes.

## Conclusion

The opportunity to replicate and extend the protocol of a previous small scale study provides confidence that our findings were robust: despite achieving substantial increases in the capacity for fat oxidation during intense exercise, 3.5 wk adaptation to a ketogenic low-CHO, high-fat diet reduced exercise economy and impaired performance of a real-life endurance event in elite athletes. In addition, this study was able to investigate (and disprove) a hypothesis based on anecdotal observations about successful performance in athletes; this is an important consideration in our current environment where testimonials and "anecdata" are given prominence. There are a number of elements identified in this study that warrant further investigation, including the health and performance benefits of longer-term adaptation to LCHF diets and a titration of exercise intensity at which the negative effects of the LCHF on exercise economy, metabolism and performance become detectable in both training and competition scenarios, thus differentiating the real-life sporting events and athletes for which this represents an unsuitable vs potentially useful practice. The potential models involving periodisation of CHO availability, or alternatively, the integration of high CHO availability within a background of keto-adaptation are numerous, and also merit investigation. The value of specific strategies of periodization of CHO availability in promoting greater training adaptations in elite athletes also remains unclear.

## Supporting information

**S1 Table. The raw data collected in this study are available in spreadsheet 1.**
(XLSX)

## Acknowledgments

Studies of this magnitude require incredible support from a large team. We thank our research colleagues and supporters: Laura Garvican-Lewis, Amelia Carr, Brent Vallance, Rita Civil, Alice Wallett, Nicolin Tee, Victor Vuong, Nikki Jeacocke, Reid Reale, Chris Fonda, Rebekah Alcock, Michelle Minehan, Bronwen Charlesson, Bronwen Lundy, Mark Howard, Toni Franklin, Ben Parker, Aki Kawamura, Susan Boegman, Kiara Carmody, Rachel McCormick, Ned Brophy-Williams, Peter Peeling, Philo Saunders, Damon Arezzollo, Jamie Whitfield, Sarah Taylor, Sacha Tee, James Tee, Stephan Praet, Claire Tallent, Phil Reading and Lynne Mercer. The support and environment of the Australian Institute of Sport were crucial in allowing this study to be completed. Finally, the generosity and commitment to sports science of the Supernova race walkers continues to inspire us. This study would not have happened without their blood, sweat, tears and good humour.

## Author Contributions

**Conceptualization:** Louise M. Burke, Avish P. Sharma, Ida A. Heikura, Sara F. Forbes, Alannah K. A. McKay, Marijke Welvaert, Megan L. Ross.

**Data curation:** Louise M. Burke, Avish P. Sharma, Ida A. Heikura, Sara F. Forbes, Melissa Holloway, Alannah K. A. McKay, Julia L. Bone, Jill J. Leckey, Marijke Welvaert, Megan L. Ross.

**Formal analysis:** Louise M. Burke, Avish P. Sharma, Ida A. Heikura, Sara F. Forbes, Alannah K. A. McKay, Marijke Welvaert, Megan L. Ross.

**Funding acquisition:** Louise M. Burke.

**Investigation:** Louise M. Burke, Avish P. Sharma, Ida A. Heikura, Sara F. Forbes, Melissa Holloway, Alannah K. A. McKay, Julia L. Bone, Jill J. Leckey, Megan L. Ross.

**Methodology:** Louise M. Burke, Avish P. Sharma, Ida A. Heikura, Sara F. Forbes, Alannah K. A. McKay, Megan L. Ross.

**Project administration:** Louise M. Burke.

**Writing – original draft:** Louise M. Burke, Avish P. Sharma, Ida A. Heikura, Sara F. Forbes, Melissa Holloway, Alannah K. A. McKay, Julia L. Bone, Jill J. Leckey, Marijke Welvaert, Megan L. Ross.

**Writing – review & editing:** Louise M. Burke, Avish P. Sharma, Ida A. Heikura, Sara F. Forbes, Melissa Holloway, Alannah K. A. McKay, Julia L. Bone, Jill J. Leckey, Marijke Welvaert, Megan L. Ross.

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
