## [Decision Letter · Decision Letter 0]

30 Sep 2019

PONE-D-19-19340

Crisis of confidence averted: Impairment of exercise economy and performance in elite race walkers by ketogenic Low Carbohydrate, High Fat (LCHF) diet is reproducible

PLOS ONE

Dear Burke,

Thank you for submitting your manuscript to PLOS ONE. After careful consideration, we feel that it has merit but does not fully meet PLOS ONE’s publication criteria as it currently stands. Therefore, we invite you to submit a revised version of the manuscript that addresses the points raised during the review process.

Address all the criticism raised by the reviewers, with special attention to the statistical analysis in whch several critical weaknesses have been spotted. Consequently the results and discussion section needs overall revision. 

We would appreciate receiving your revised manuscript by October 20th. To enhance the reproducibility of your results, we recommend that if applicable you deposit your laboratory protocols in protocols.io, where a protocol can be assigned its own identifier (DOI) such that it can be cited independently in the future. For instructions see: http://journals.plos.org/plosone/s/submission-guidelines#loc-laboratory-protocols

We look forward to receiving your revised manuscript.

Kind regards,

Andrea Martinuzzi

Academic Editor

PLOS ONE

Journal Requirements:

1. We note that you have included the phrase “data not shown” in your manuscript. Unfortunately, this does not meet our data sharing requirements. PLOS does not permit references to inaccessible data. We require that authors provide all relevant data within the paper, Supporting Information files, or in an acceptable, public repository. Please add a citation to support this phrase or upload the data that corresponds with these findings to a stable repository (such as Figshare or Dryad) and provide and URLs, DOIs, or accession numbers that may be used to access these data. Or, if the data are not a core part of the research being presented in your study, we ask that you remove the phrase that refers to these data.

2. We note you have included a table to which you do not refer in the text of your manuscript. Please ensure that you refer to Table 4 in your text; if accepted, production will need this reference to link the reader to the Table.

Reviewers' comments:

Reviewer's Responses to Questions

**Comments to the Author**

1. Is the manuscript technically sound, and do the data support the conclusions?

Reviewer #1: Partly

Reviewer #2: Yes

Reviewer #3: Yes

2. Has the statistical analysis been performed appropriately and rigorously? 

Reviewer #1: No

Reviewer #2: Yes

Reviewer #3: Yes

3. Have the authors made all data underlying the findings in their manuscript fully available?

Reviewer #1: Yes

Reviewer #2: Yes

Reviewer #3: Yes

4. Is the manuscript presented in an intelligible fashion and written in standard English?

Reviewer #1: Yes

Reviewer #2: Yes

Reviewer #3: Yes

5. Review Comments to the Author

Reviewer #1: Abstract

- when reporting effects attributed to the diets, please report the estimate of effect followed by the confidence intervals and p-values; reporting p-values and confidence intervals only as you have done in line 42 is not informative.

- avoid using abbreviations that are not previously introduced, e.g. VO2peak in line 42.

- include numerical results to back up the cited results; there are no numerical results to illustrate the increased oxygen cost of walking at race-relevant velocities in line 44.

- it is unclear what "with a trend for faster time" in line 46 means; in any case, the numerical results presented to not indicate statistical evidence of a difference in whatever comparison is being made, to justify such a definitive conclusion here.

Introduction

- it is incorrect to suggest, as has been done in Table 1, that the use of mixed modelling was sufficient to account for the inherent bias in a non-randomised study, especially one where group assignment is done according to participants own beliefs about the intervention. At best, mixed modelling, as with any other approaches to regression, can help deal with observed imbalances; however, non-randomised trials are prone to unobserved bias-inducing imbalances which no approach to the statistical analysis can ameliorate. The authors should clearly acknowledge this and factor it into all interpretations of the findings of the study.

Methods

- the authors should provide a detailed justification for the number of participants recruited into the various arms of the study rather than citing the size of the previous study (e.g. by providing a sample size calculation).

- this analysis entails multiple statistical testing; the authors should adopt an approach to adjust for multiple testing, and update their results and discussion to reflect these adjustments to the results.

Results

- a table of the pre-test characteristics of participants allocated to the three diets should be presented first before any comparison of outcomes across the groups. This table should indicate the means and SDs of continuous variables and counts and proportions of categorical ones. As this was a non-randomised experiment, a formal comparison (i.e. statistical tests of differences between groups, relative to one comparison group, with p-values) should also be conducted in this table to show how well the groups were balanced in terms of participant characteristics.

- table 3 is unclear; it is difficult to ascertain what comparisons are being made here. It is also not clear what the numbers indicated in +/- mean. This designation should not be used here. If the authors wish to present the nutrient intakes across the diet groups, then the different diets should be presented in columns (not separate rows); the means and standard error (SE) for each nutrient should be presented, along with differences, their confidence intervals and p-values for the differences in intake across diet groups, with one group as a reference group. Given that all participants followed the HCHO diet during De-adapt, HCHO could be used as the reference group (the authors could use any other group that allows the authors to address the hypothesis about exercise economy and performance with the LCHF diet). Separate tables for the adapt and de-adapt phases should be presented (or separate columns if one wide-enough table can be devised).

- a similar format suggested for table 3 should be used for table 4.

- Table 5 is cluttered and does not communicate the intended message. It should be updated as follows

(1) if body mass was measured pre-test, then it should be reported in the table of baseline characteristics

(2) baseline RER, VO2, HR and RPE should all be reported in the baseline table as well

(3) This table should then report "differences in differences" for each outcome. To do this the baseline-adapt differences for each diet should be reported with its standard error (not standard deviation - additionally the +/- designation should not be used); then the between-diet differences in the baseline-adapt differences, with 95% confidence intervals and p-values for those differences, should subsequently be reported. The narrative of the results should be updated to comment on these differences.

These recommendations about the presentation of results should be extended to other comparisons of outcomes between the diet groups. Effects estimated as differences in outcome between groups, or differences in differences as described above, should be presented with their confidence intervals and p-values, and not p-values alone as currently done for many comparisons, as p-values alone are not informative.

The +/- designation in the numerical results should be avoided except where the authors are trying to report and central estimate and its ranges. Means +/- SDs are meaningless for inference.

Discussion and conclusions

This study has two major problems: (1) its small size (notwithstanding the lack of a formal and objective justification of the study size ), and (2) the lack of randomisation in the study design, and the implied possibility of latent confounding, which cannot be remedied by any approach to the analysis contrary to what the authors suggest. The discussion and conclusion should acknowledge and be tempered in light of these. The reproduced previous findings is hardly an aversion of confidence in the conclusions, given these weaknesses.

Reviewer #2: Line 516: Replace "glucose" with "lactate"

This was an important study to be conducted due to the lack of evidence in general on how a ketogenic diet affects endurance exercise performance. Your previous publication is arguably the strongest evidence to date due to the strict dietary control and use of elite athletes that can help isolate the effect of the diet. Therefore, it was an excellent choice to take the opportunity to build upon your previous work through replicating the prior study and addressing some short-comings. The manuscript was well-written and addressed all important points. The statistical analysis was properly conducted and the findings were reported accurately. Excellent job.

Reviewer #3: OVERALL

This manuscript repeated a study previously done by the same group of authors. It assessed how different dietary programs, i.e. ketogenic, high carbohydrate and periodized carbohydrate, affected exercise economy and performance responses to a period of intensified exercise training. Although the dataset is largely a repeat of previous work, the authors addressed some criticisms of the initial manuscript in the current design, most notably adding a period of high carbohydrate availability between the end of intensified training and a race. The importance of repeatable data and the novel aspects to this study make this manuscript is deserving of publication in PLOS One, but some concerns must be addressed first.

MAJOR REVISIONS

[1] P-value reporting is challenging to interpret at times and sometimes p-values are not reported. Please include a p-value for each comparison mentioned even if not significant.

[2] Statistical analysis – reporting and method selection.

a. First, can you please explain your statistical analyses in greater detail here? To my understanding, a general linear mixed model is commonly used to predict something based on fixed and random effects. Due to this, I’m unsure how this would detect differences in time trial duration. It makes sense to me that you have entered diet and base/adapt as fixed effects and subject as random, but it seems like these would predict an outcome variable instead of test for differences between groups. Additionally, please justify why you feel this methodology is appropriate despite the relatively small sample size used.

b. Second, why did you select this method as opposed to an ANCOVA, where the covariate is subject training status or baseline time trial duration? I like that you are trying to control for baseline differences between subjects, however an ANCOVA may be a more appropriate method to achieve what you are trying to when considering that mixed models are typically used for very large sample sizes. It may also increase your chances of finding a difference between diet interventions at Adapt.

c. Third, when building this model why not include sex as a fixed factor? You have acknowledged it may affect you results but don’t include it in the model. Also, please list all fixed and random effects initially entered into the model and then which you decided to include and exclude. If you decide to keep with this analysis it is important to include this information for repeatability purposes.

d. Last, please include effect sizes for the physiological and performance effects. At the very least, it would be useful to add a Cohen’s dz or d for comparisons between two groups

[3] Graphical representation of data. Showing individual data points on your graph, or at least plots that show more data than mean + SD, are recommended. It would be nice to see individual data points that are connected for repeated measures, i.e. Figure 3 showing a line between Baseline and Adapt within each diet that connects the two points for each subject. Although significant statistically it is still informative, especially considering the relatively small N, to show the individual responses. I feel that this is particularly important for Figures 3 and 5 (economy and performance data), and less important and likely not necessary for metabolite data.

[4] There is a lot of writing throughout that seems to not be essential for a scientific audience, particularly in the discussion. For example, explanations of scientific processes (i.e. L669-674). Please consider condensing. Likewise, the nitrate-nitrite-NO reduction pathway discussion and relevance to BRJ responses in the discussion, although somewhat relevant, do not directly relate to the data set. The authors should consider only elaborating on points that directly relate to their dataset, and it is recommended that they spend more time talking about PCHO results instead.

MINOR REVISIONS

Abstract

L51. Consider another phase instead of keto-adaptation. Adaptation refers to changes over generations within a species; and therefore, adaptations would not have occurred in this study. Please revise throughout or provide an explanation as to why this is the most appropriate descriptor.

Introduction

The introduction was good for LCHF and HCHO groups, however it did not effectively introduce the PCHO group. Please include some more background on and rationale for having a PCHO group.

Methods

Were any of the subjects who participated in this study also in your previous study? Particularly anyone in the LCHF group. Considering you are trying to replicate results, it is important for the reader to know if this is a completely, or partially, different subject group.

L167. Is there a more appropriate word than “succumbed” to describe what happened?

L299. Do you have any data to prove that athletes are indeed at steady state during this time?

L306. Did you take metabolite measurements in single, duplicate, triplicate? Please specify.

L306. The analyzer you used measures D-b-hydroxybutyrate only. Since, it does not measure other ketone bodies (acetone and acetoacetate) it is incorrect to say “ketones”. Please change the label to beta-hydroxybutyrate throughout.

L309. This is a good control method. Do you have any data/evidence to support inter-device variability?

L313. Please specify that these are estimated rates of “whole-body” CHO and fat oxidation.

L318-323. Regarding not including ketone oxidation to VO2 and RER. I agree that you should not have ketone body oxidation in your calculations because you don’t have the data to calculate it without making significant stretches. Furthermore, some of your “ketone” data show it increases pre- to post-exercise (L520); and therefore, if using the same calculations as Cox et al. 2016 Cell Metab, you would get a negative contribution of “ketone bodies” to VO2. My concern, however, is the error from not including this is not necessarily systematic because you have a training program and exercise training has been implicated with an increased ability of skeletal muscle to oxidize ketone bodies (For review see Evans et al. 2017, J Phys). Thus, the error obtained from excluding ketone bodies in your energy expenditure estimations may be larger post- vs. pre-training.

L348. Why not have everyone consume a HCHO pre-race? Based on the differences you found when comparing Base vs. Adapt within diet treatments and the non-significant difference in 20 km TT performance at the end, the data set cannot distinguish whether the impaired performance on LCHF was due to a single LCHF meal pre-race or chronic LCHF intake during training. Also, when comparing LCHF at Adapt to Baseline, the pre-race meals would have been different because you said all were on a HCHO diet at Baseline. This should receive more attention in the discussion. Please advise if I have misunderstood the pre-race diet data.

Power calculation. You performed an a priori power calculation, however, did not include enough data to make it repeatable. Please include the effect size, alpha, power, statistical test, etc. that were input into your calculation and state the outcome variable the calculation was performed on.

L372. Can you clarify what you mean by “random intercept for the athletes”. Do you mean a certain characteristic of athletes, i.e. aerobic fitness or time-trial (TT) duration at Baseline, or is it simply a random intercept for “subject”? Also, would it be better to include Baseline or personal best TT duration when assessing differences in TT data at Adapt instead of aerobic fitness?

It is good you checked normality with a QQ plot, however the authors are recommended to also include an objective normality test to compliment this subjective normality test.

L382. Why did you use Bonferroni post-hoc test as opposed to the more conventional Tukey’s HSD?

L384. Please report which variables did not pass the normality test.

How was normality assessed for diet intake data? This should be added to the manuscript.

L385. When analyzing diet, you ignored macronutrient composition in your ANOVA/t-test. I.e., to my understanding, diet was analyzed with a t-test (group) within a macronutrient vs. 2-way ANOVA (group x macronutrient). Does analyzing the data this way change the story?

L543. How did you normalize these data? I.e. relative to race 1 performance?

Please explain why effect sizes were not computed for this study, particularly for performance and economy data.

Results

Do you have Wpeak data? It would be interesting to see if LCHF had a higher Wpeak in concert with a higher VO2peak, or if the higher VO2peak was simply a result of reduced BM and/or increased fat oxidation.

L419. Specify what groups each value belongs to.

L450. Specify if these changes were significant in addition to being “minor”.

L454. Did the comparison of VO2 significantly differ between diets at Adapt?

L463-468. Why is your economy data shown as relative VO2? The LCHF diet group lost BM and therefore irrespective of changes in economy would report higher relative VO2s than the other two groups regardless of changes in economy. It would be best to show this data as absolute VO2, or even better, use BM as a covariate in your analysis.

L583. The fact 3 participants got a personal best in LCHF/HCHO makes for an interesting talking point. Please add this to the discussion.

Discussion

Overall the discussion spent a lot of time on data that you had already shown in Burke et al. 2017. I recommend that you shorten this section as what is being discussed isn’t novel per se. It is more important to talk about, as you have highlighted in the intro, how it was repeatable.

L556. Do you have a reference to demonstrate that the intensities you assessed are similar to those that an ultra-endurance athlete competes at?

L656. The way this is worded makes it seem like HCHO and PCHO diets improved performance and LCHF decreased performance, although this is not that you found. Please consider rewording to better reflect that PCHO saw no change in performance.

L638. Do you have any insight into why blood ketones did not increase as much in this study (Figure 3, ~0.8 mM) vs. your previous work (~1.5 mM at same point)? Looking at absolute values there appears to be a slight difference in means.

L641. Some may argue that glycolytic activity was not the same despite similar blood lactates. A LCHF diet reduced PDH flux and transformation was more in an “inactive” state, which you mention later. Therefore, for the same pyruvate generation in LCHF vs. HCHO, you’d expect more lactate production in HCHO. Also lactate clearance from blood or muscle could have been affected, or lactate generation; or glycogenolysis could have been reduced. The authors go into more detail later, but should list some alternate explanations for blood lactate results in addition to the speculation they have proposed.

L661. Similar to my comment on ketone oxidation and training above, this error may not be systematic. Please address this.

L666. Including an effect size would help objectively make your point of a small but critical change.

L694. Including individual data or an effect size such as Cohen’s dz would make this point significantly stronger. Also, having ES for this study would make comparisons to your previous work stronger.

708. Did you account for differences in training quality and volume in your statistical model? How do we know the changes in TT performance aren’t because of differences in training volume reported in the first half of the program?

L734. Can you list any more muscle-specific consequences of this in addition to BP? The discussion mentions many muscle-specific effects of LCHF and including a muscle-specific effect here would complement the discussion nicely.

L775. Please clarify what you mean by and how you “removed the independent…”. Do you mean these were included in your statistical model?

L782. You should consider referencing Evans et al. 2018 and 2019 in MSSE in addition to Leckey et al. to make your argument stronger. These papers do not show an improvement in performance despite using the same supplement as ref 71 (Cox et al.), which showed a performance improvement.

L792-798. Similar to my main discussion comment this is a part of the discussion that seems unnecessary for a scientific audience and could be more concisely summarized by citing a review. Please consider shortening.

L808. Reference 78 is not in the reference list.

L832. Instead of chronic, consider stating duration in weeks as some people would argue that your exposure to LCHF was not “chronic”.

Tables and Figures

Figures 3 and 5. Please consider using individual data (as per above).

Figure 4. Change y-axis on 3C from “ketones” to “D-beta-hydroxybutyrate”.

Figure 5 B and C. Is there a graphical method to represent what you have described in your results section for this (L542-552). This is a really interesting statement and this method of analysis teases out a difference in a real-world time trial between diets. Also, please consider placing the 20 km race means for both diets beside each other to allow for easier comparisons. Additionally, if you are going to show individual data points here, it may be useful to assign each subject a symbol throughout to best show your repeated measures data because lines connecting 3 time points gets messy.

6. PLOS authors have the option to publish the peer review history of their article (what does this mean?). If published, this will include your full peer review and any attached files.

Reviewer #1: No

Reviewer #2: Yes: Alexander Leaf

Reviewer #3: Yes: Devin G McCarthy

---

## [Author Response · Author response to Decision Letter 0]

5 Mar 2020

OVERALL

This manuscript repeated a study previously done by the same group of authors. It assessed how different dietary programs, i.e. ketogenic, high carbohydrate and periodized carbohydrate, affected exercise economy and performance responses to a period of intensified exercise training. Although the dataset is largely a repeat of previous work, the authors addressed some criticisms of the initial manuscript in the current design, most notably adding a period of high carbohydrate availability between the end of intensified training and a race. The importance of repeatable data and the novel aspects to this study make this manuscript is deserving of publication in PLOS One, but some concerns must be addressed first. 

Thank you for your kind summary. We feel that replication studies are important, and that we have been able to achieve this as well as add new components. We have incorporated most of your suggestions (or addressed concerns) to enhance the final MS. Thank you for the advice.

MAJOR REVISIONS

[1] P-value reporting is challenging to interpret at times and sometimes p-values are not reported. Please include a p-value for each comparison mentioned even if not significant. 

P values have been added as requested

[2] Statistical analysis – reporting and method selection. 

The statistician within our team (Marijke Welvaert) has addressed the queries below. In addition to explaining the choice of statistical methods for the current paper, we point out that the replication aspect of this investigation required us to repeat the same analytical methods

First, can you please explain your statistical analyses in greater detail here? To my understanding, a general linear mixed model is commonly used to predict something based on fixed and random effects. Due to this, I’m unsure how this would detect differences in time trial duration. It makes sense to me that you have entered diet and base/adapt as fixed effects and subject as random, but it seems like these would predict an outcome variable instead of test for differences between groups. Additionally, please justify why you feel this methodology is appropriate despite the relatively small sample size used.

While a general mixed model can indeed be used for predictions based on both fixed and random effects, however, test statistics can be calculated from the marginal models to look at the fixed effects more specifically and investigate group sizes. The linear mixed model has been shown to be robust under small samples and due to its maximum likelihood estimation allows for using all available data as opposed to a complete case analysis for a RM-ANOVA. Furthermore, we used Kenward-Roger Type II F tests which were specifically developed for small samples (Kenward, M. G., Roger, J. H. (1997). Small sample inference for fixed effects from restricted maximum likelihood. Biometrics 53:983–997)

a. Second, why did you select this method as opposed to an ANCOVA, where the covariate is subject training status or baseline time trial duration? I like that you are trying to control for baseline differences between subjects, however an ANCOVA may be a more appropriate method to achieve what you are trying to when considering that mixed models are typically used for very large sample sizes. It may also increase your chances of finding a difference between diet interventions at Adapt.

Whilst there is no statistical literature directly comparing the suggested ANCOVA approach and the linear mixed model, the power should be equivalent in the case when there are only 2 time points. However, when comparing multiple time points this is no longer possible following the ANCOVA approach as this would violate the independence assumption. Furthermore, ANCOVA uses the same asymptotic properties as linear mixed model for inference, and as discussed before, the linear mixed model has been shown to also be robust in small samples and has its applications far beyond large samples.

b. Third, when building this model why not include sex as a fixed factor? You have acknowledged it may affect you results but don’t include it in the model. Also, please list all fixed and random effects initially entered into the model and then which you decided to include and exclude. If you decide to keep with this analysis it is important to include this information for repeatability purposes.

This is described in the methods. Only non-significant interactions were dropped from the model, but all fixed effects were always included. We did not include sex however. This would even further decrease the group sizes given that there were only 1-3 females in each treatment. Any differences due to gender would be absorbed through the baseline adjustment, but cannot be explicitly tested without it being in the model.

c. Last, please include effect sizes for the physiological and performance effects. At the very least, it would be useful to add a Cohen’s dz or d for comparisons between two groups

These have been added throughout the MS

[3] Graphical representation of data. Showing individual data points on your graph, or at least plots that show more data than mean + SD, are recommended. It would be nice to see individual data points that are connected for repeated measures, i.e. Figure 3 showing a line between Baseline and Adapt within each diet that connects the two points for each subject. Although significant statistically it is still informative, especially considering the relatively small N, to show the individual responses. I feel that this is particularly important for Figures 3 and 5 (economy and performance data), and less important and likely not necessary for metabolite data.

We have done this for figure 5 since it tells the story well. However, despite many attempts we were unable to draw a figure 3 that was legible with all the individual points noted. At the end of the day, the performance effects are of the greatest interest in this area of work, so we have redrawn the figures to make this the headline story 

[4] There is a lot of writing throughout that seems to not be essential for a scientific audience, particularly in the discussion. For example, explanations of scientific processes (i.e. L669-674). Please consider condensing. Likewise, the nitrate-nitrite-NO reduction pathway discussion and relevance to BRJ responses in the discussion, although somewhat relevant, do not directly relate to the data set. The authors should consider only elaborating on points that directly relate to their dataset, and it is recommended that they spend more time talking about PCHO results instead.

MINOR REVISIONS

Abstract

L51. Consider another phase instead of keto-adaptation. Adaptation refers to changes over generations within a species; and therefore, adaptations would not have occurred in this study. Please revise throughout or provide an explanation as to why this is the most appropriate descriptor.

Keto-adaptation is the term that is used in this area of nutrition (both sport and health-related), and is widely used in both scientific and lay circles. Since this is what those even vaguely familiar with the LCHF diet understand, it is the best term to use. It will be understood by those reading the MS, and as a keyword, it will help the paper to be tagged/searched by the widest audience possible

Introduction

The introduction was good for LCHF and HCHO groups, however it did not effectively introduce the PCHO group. Please include some more background on and rationale for having a PCHO group.

We have included a small amount of additional information there to better represent the PCHO concept, leaving the discussion section for a greater summary of the current knowledge of this dietary principle, placing our results in better context

Methods

Were any of the subjects who participated in this study also in your previous study? Particularly anyone in the LCHF group. Considering you are trying to replicate results, it is important for the reader to know if this is a completely, or partially, different subject group.

There was a small amount of overlap between the two study cohorts – this was explained already in the discussion (second paragraph). Six of the subjects in the current cohort had participated in the previous study and two of these chose to do the LCHF diet again.

L167. Is there a more appropriate word than “succumbed” to describe what happened?

We have replaced this with “incurred”

L299. Do you have any data to prove that athletes are indeed at steady state during this time?

This laboratory protocol is based on established best practice guidelines of the Australian Institute of Sport (Tanner and Gore, Physiological tests for elite athletes, 2nd edition, Human Kinetics, 2013). It has been undertaken tens of thousands of times with elite endurance athletes in our laboratory. Although we would normally implement 4 minute stages for this test, we typically see an achievement of “steady state”, as shown by stable readings for the pulmonary gas characteristics within 90-120 s of duration of the new intensity/workload. We were confident to reduce the stages to 3 minutes to allow us to undertake the testing of all participants within 2 days. The benefits of running this protocol as a single training camp have already been established, but require clever timetabling. 

L306. Did you take metabolite measurements in single, duplicate, triplicate? Please specify.

The measurements were taken as single samples and analysed immediately. If there was a discrepancy in the reading between actual and what might be expected (very occasionally), the test was repeated

 306. The analyzer you used measures D-b-hydroxybutyrate only. Since, it does not measure other ketone bodies (acetone and acetoacetate) it is incorrect to say “ketones”. Please change the label to beta-hydroxybutyrate throughout. 

This has been done

L309. This is a good control method. Do you have any data/evidence to support inter-device variability?

We did some pilot testing that suggested small inter-device differences, but nothing that could be cited as definite proof of the inter-device difference 

L313. Please specify that these are estimated rates of “whole-body” CHO and fat oxidation.

This has been done

L318-323. Regarding not including ketone oxidation to VO2 and RER. I agree that you should not have ketone body oxidation in your calculations because you don’t have the data to calculate it without making significant stretches. Furthermore, some of your “ketone” data show it increases pre- to post-exercise (L520); and therefore, if using the same calculations as Cox et al. 2016 Cell Metab, you would get a negative contribution of “ketone bodies” to VO2. My concern, however, is the error from not including this is not necessarily systematic because you have a training program and exercise training has been implicated with an increased ability of skeletal muscle to oxidize ketone bodies (For review see Evans et al. 2017, J Phys). Thus, the error obtained from excluding ketone bodies in your energy expenditure estimations may be larger post- vs. pre-training. 

This has been done in the discussion

L348. Why not have everyone consume a HCHO pre-race? Based on the differences you found when comparing Base vs. Adapt within diet treatments and the non-significant difference in 20 km TT performance at the end, the data set cannot distinguish whether the impaired performance on LCHF was due to a single LCHF meal pre-race or chronic LCHF intake during training. Also, when comparing LCHF at Adapt to Baseline, the pre-race meals would have been different because you said all were on a HCHO diet at Baseline. This should receive more attention in the discussion. Please advise if I have misunderstood the pre-race diet data.

The goal of our studies was to investigate the effect of the much-touted ketogenic LCHF diet on performance of elite endurance athletes. This meant we needed to implement the intervention as it has been devised – i.e. as implemented in the original study (Phinney et al., 1983) and as presented in the Volek/Phinney book (and many many testimonials). The principle of keto-adaptation is that by chronically consuming a LCHF diet, the athlete is able to increase the contribution of fat to muscle fuel needs so that they no longer need to consume CHO around exercise sessions. Therefore, the intervention for the LCHF group in the Adapt race required a carbohydrate-restricted pre-race meal to follow the protocol as an intact implementation; the outcry from the LCHF community had we departed from the LCHF diet characteristics would have been deafening! At one level, it is not possible to isolate the effect of the acute (race day) nutritional interventions from those of the chronic (3.5 week) strategies. However, the “keto” diet is a total dietary change that involves acute plus chronic elements. There is obviously further opportunity for mixing and matching the various elements of chronic and race day feeding in future studies, but I feel that we have clearly justified the need to investigate the LCHF treatment as it has been described (and previously investigated in a non-competitive scenario). As a replication study, it needed also to duplicate the treatments used in the first study.

The dietary strategies for the 20 km race represented a “real life” investigation of the claims from observers of our first study that individuals who had undertaken the LCHF diet went on to “breakthrough” performances in the weeks after their camp involvement. We were able to directly test this hypothesis by recreating their experience, but also comparing it to the outcomes of athletes who had undertaken the same “training camp experience” supported by a carb-rich diet throughout. Again, there are various permutations and combinations of acute and chronic feeding strategies, but we have justified the case for the examination of the protocol we implemented. 

Power calculation. You performed an a priori power calculation, however, did not include enough data to make it repeatable. Please include the effect size, alpha, power, statistical test, etc. that were input into your calculation and state the outcome variable the calculation was performed on. 

We did not undertake a traditional power calculation for this study. We based the sample size requirements for this study on a combination of 1. past experience from a large number of other investigations where 8-10 subjects have been sufficient to detect a performance change of 2-3% which is usually both worthwhile and statistically significant, 2. The availability of world class athletes who were willing to undertake the interventions/testing and 3. The resources and experience of the research team in being able to manage rigorous control of the intervention and testing. The fact that we have achieved significant changes in metabolic parameters and performance is not just “good luck”; it builds on 30 years of experience of investigations of various training interventions in high performance athletes.

L372. Can you clarify what you mean by “random intercept for the athletes”. Do you mean a certain characteristic of athletes, i.e. aerobic fitness or time-trial (TT) duration at Baseline, or is it simply a random intercept for “subject”? Also, would it be better to include Baseline or personal best TT duration when assessing differences in TT data at Adapt instead of aerobic fitness?

Yes, the random intercept for the athletes refers to a random intercept for subject, which in this context are athletes. The random intercept allows for each participant to have their own starting point at baseline, therefore any differences compared to baseline are adjusted for their personal fitness level/performance.

It is good you checked normality with a QQ plot, however the authors are recommended to also include an objective normality test to compliment this subjective normality test.

We understand there is some subjectivity in the normality test but it has been demonstrated that the linear mixed model is quite robust against diversions from normality (Hélène Jacqmin-Gadda, Solenne Sibillot, Cécile Proust, Jean-Michel Molina, Rodolphe Thiébaut, Robustness of the linear mixed model to misspecified error distribution, Computational Statistics & Data Analysis, Volume 51, Issue 10, 2007, Pages 5142-5154). 

L382. Why did you use Bonferroni post-hoc test as opposed to the more conventional Tukey’s HSD?

We have re-run the stats using Tukey's test as a posthoc - this has also been revised on line 384

L384. Please report which variables did not pass the normality test.

These have been reported on L382

How was normality assessed for diet intake data? This should be added to the manuscript.

This information has been added on L 381

L385. When analyzing diet, you ignored macronutrient composition in your ANOVA/t-test. I.e., to my understanding, diet was analyzed with a t-test (group) within a macronutrient vs. 2-way ANOVA (group x macronutrient). Does analyzing the data this way change the story?

All the macronutrient data shown on tables 3 and 4 has been included in the two-way ANOVA (table 3: group x phase) or t-tests (table 4; between diets) 

L543. How did you normalize these data? I.e. relative to race 1 performance?

By “Normalising” data, we mean that we considered the 20 km race to be twice as long as the 10,000 race, and compared the outcomes of this race to both of the 10,000 races (i.e. was it different than could have been predicted from either of the 10,000 m races)

Please explain why effect sizes were not computed for this study, particularly for performance and economy data.

These have now been added

Results

Do you have Wpeak data? It would be interesting to see if LCHF had a higher Wpeak in concert with a higher VO2peak, or if the higher VO2peak was simply a result of reduced BM and/or increased fat oxidation. 

We don’t have Wpeak data. We agree that changes in absolute power would be amplified by the changes in body composition and have commented about that in relation to changes in VO2max 

L419. Specify what groups each value belongs to.

This has now been added

L450. Specify if these changes were significant in addition to being “minor”.

This has now been added

L454. Did the comparison of VO2 significantly differ between diets at Adapt?

This comparison between diets is not of practical relevance; as discussed below, in terms of performance in a weight-bearing sport, oxygen cost is best considered relative to the mass that must be moved and the maximal capacity of oxygen see below.

L463-468. Why is your economy data shown as relative VO2? The LCHF diet group lost BM and therefore irrespective of changes in economy would report higher relative VO2s than the other two groups regardless of changes in economy. It would be best to show this data as absolute VO2, or even better, use BM as a covariate in your analysis.

The economy data, expressed in absolute L of oxygen, has been presented in table 5 and received brief commentary. In terms of what is relevant and well known in high performance endurance sport (at least weight bearing sports like running and race walking), economy is best discussed in terms of the oxygen cost of moving the athlete’s body mass and how this relates to their VO2max. This terminology takes into account the total number of changes that occur during the treatment (changes in mass and aerobic capacity) and best represents important performance characteristics – the energetic demand of moving the athlete over the race distance and how much this costs them in terms of their overall capacity. Athletes and coaches, and the sports scientists who work with them, know what the numbers mean in absolute terms and in terms of the changes that might be achieved by various interventions. For example, the advantage of the Nike Zoomfly elite 4% shoes has been examined by comparing the relative oxygen cost (ml/kg/min) at different running speeds (Barnes and Kilding 2019). Furthermore, the coach attached to our study, who has coached several athletes to Olympic and World Championship medals has decades of unpublished data that show that male race walkers who can walk at 13 km/hr during the current economy protocol (second speed of our test) at less than 70%VO2max are highly likely to be internationally successful. 

We have added a brief section around why these numbers are important to the discussion section rather than comment on them here

L583. The fact 3 participants got a personal best in LCHF/HCHO makes for an interesting talking point. Please add this to the discussion.

I am not sure of the talking point here. We have added to the discussion that our study involved elite athletes who competed in real life races with the usual rewards of their occupations (prize money, official recognition, qualification times for international events). It is entirely expected that some would achieve their career trajectories/goals of improving from past performances, particularly as a result of training in a high performance environment for 6 weeks. There were approximately equal numbers from both treatment groups who were able to achieve a personal best in the 20 km race. This speaks to the authenticity of our study – that we were able to capture the phenomena that elite athletes continue to refine their training to achieve incremental improvements in performance over their careers, and when sufficiently prepared and motivated, can achieve personal bests. Apart from supporting the benefits of the “training camp effect”, we were able to confirm that a 3 week restoration of training with a high carbohydrate diet was likely to overturn the previous performance impairment seen in the second of the 10,000 m races. However, there was no indication that this group received an advantage over the cohort who trained with high carbohydrate availability for the whole of the 6 week period as had been proposed from the testimonials/observations of the first study. The sports world is filled with “anecdata” – where the strategies used by successful athletes are given credit for successful performances. Our study is unique in providing a controlled investigation of one example of such anecdata.

Discussion

Overall the discussion spent a lot of time on data that you had already shown in Burke et al. 2017. I recommend that you shorten this section as what is being discussed isn’t novel per se. It is more important to talk about, as you have highlighted in the intro, how it was repeatable.

We have tried to change the discussion to remove some of the repetition from the first paper, and replace it with novel insights (gleaned from your questions/comments above) and discussions about the repeatability. We have also been able to add some new data from other labs which confirms our findings and speculated mechanisms.

L556. Do you have a reference to demonstrate that the intensities you assessed are similar to those that an ultra-endurance athlete competes at?

There are few data that document the exercise intensities at which elite athletes compete. We have noted in the study in several places that the speeds used in the economy test relate to those of race pace for race walking. However we have also added some references that discuss race intensities for other sports.

L656. The way this is worded makes it seem like HCHO and PCHO diets improved performance and LCHF decreased performance, although this is not that you found. Please consider rewording to better reflect that PCHO saw no change in performance.

This has been done

L638. Do you have any insight into why blood ketones did not increase as much in this study (Figure 3, ~0.8 mM) vs. your previous work (~1.5 mM at same point)? Looking at absolute values there appears to be a slight difference in means.

We haven’t done a direct comparison but we note some individual variability so it is difficult to justify a commentary on this. We feel that the results of the present study show that the LCHF diet was successfully implemented in that it achieved blood concentrations of BHB that meet the definitions provided by keto diet advocates.

L641. Some may argue that glycolytic activity was not the same despite similar blood lactates. A LCHF diet reduced PDH flux and transformation was more in an “inactive” state, which you mention later. Therefore, for the same pyruvate generation in LCHF vs. HCHO, you’d expect more lactate production in HCHO. Also lactate clearance from blood or muscle could have been affected, or lactate generation; or glycogenolysis could have been reduced. The authors go into more detail later, but should list some alternate explanations for blood lactate results in addition to the speculation they have proposed. 

This is a good point but we have removed the discussion of changes in carb and fat metabolism from the text - this meets the suggestions that we remove elements that have been previously discussed in the earlier study but lack new data to expand on in this paper.

L661. Similar to my comment on ketone oxidation and training above, this error may not be systematic. Please address this.

This is a good point and has been addressed in the revised MS

L666. Including an effect size would help objectively make your point of a small but critical change.

This is a good point and we have addressed this in the revised MS by comparing the change to that seen with the new running shoes – this is a major talking point in endurance sport at present so it is even more impactful that a statistical term

L694. Including individual data or an effect size such as Cohen’s dz would make this point significantly stronger. Also, having ES for this study would make comparisons to your previous work stronger.

ES have been added to our results and we have now tried to discuss the real world significance of our findings

708. Did you account for differences in training quality and volume in your statistical model? How do we know the changes in TT performance aren’t because of differences in training volume reported in the first half of the program? 

We cannot rule out that small changes in training quality in the first half of the program might have contributed to the race results. We had already included this in our discussion. However, we still feel that the major finding of the change in economy has a more global and significant effect on performance.

L734. Can you list any more muscle-specific consequences of this in addition to BP? The discussion mentions many muscle-specific effects of LCHF and including a muscle-specific effect here would complement the discussion nicely.

We are not aware of studies that have investigated this. The role of nitrate-NO pathway in exercise capacity is new, and the current focus is on changes to enhance pathway via nitrate supplementation

L775. Please clarify what you mean by and how you “removed the independent…”. Do you mean these were included in your statistical model?

We mean that we have avoided the introduction of large changes in BM (e.g. ~ 6 kg) as have occurred in some other studies of the LCHF diet in athletes where this has a separate effect on performance. The term “separate” is better than “independent” in explaining this, and we have rewritten the sentence to make it clearer.

L782. You should consider referencing Evans et al. 2018 and 2019 in MSSE in addition to Leckey et al. to make your argument stronger. These papers do not show an improvement in performance despite using the same supplement as ref 71 (Cox et al.), which showed a performance improvement.

Good point – we have amended the text to include these papers

L792-798. Similar to my main discussion comment this is a part of the discussion that seems unnecessary for a scientific audience and could be more concisely summarized by citing a review. Please consider shortening.

We are confused by this comment since previous directions have been to increase the commentary around the PCHO diet. Furthermore, although we recognise the readership of the journal is a scientific audience, we feel that not all will be dedicated to applied sport science and might not be fully aware of the current interest in periodising CHO availability.

L808. Reference 78 is not in the reference list.

This has been added – thanks for pointing out the omission.

L832. Instead of chronic, consider stating duration in weeks as some people would argue that your exposure to LCHF was not “chronic”.

Good point – we have amended the text

Tables and Figures

Figures 3 and 5. Please consider using individual data (as per above).

We have been able to do this for figure 5 so that it enhances the interpretation of the outcome. 

Figure 4. Change y-axis on 3C from “ketones” to “D-beta-hydroxybutyrate”.

This has been done

Figure 5 B and C. Is there a graphical method to represent what you have described in your results section for this (L542-552). This is a really interesting statement and this method of analysis teases out a difference in a real-world time trial between diets. Also, please consider placing the 20 km race means for both diets beside each other to allow for easier comparisons. Additionally, if you are going to show individual data points here, it may be useful to assign each subject a symbol throughout to best show your repeated measures data because lines connecting 3 time points gets messy. 

We think the new figure 5 achieves your suggestion

---

## [Decision Letter · Decision Letter 1]

20 Apr 2020

PONE-D-19-19340R1

Crisis of confidence averted: Impairment of exercise economy and performance in elite race walkers by ketogenic Low Carbohydrate, High Fat (LCHF) diet is reproducible

PLOS ONE

Dear Dr. Burke,

Thank you for submitting your manuscript to PLOS ONE. After careful consideration, we feel that it has merit but does not fully meet PLOS ONE’s publication criteria as it currently stands. Therefore, we invite you to submit a revised version of the manuscript that addresses the points raised during the review process.

Please address the emaining minor observations by reviewer 3.

We would appreciate receiving your revised manuscript by April 24th. To enhance the reproducibility of your results, we recommend that if applicable you deposit your laboratory protocols in protocols.io, where a protocol can be assigned its own identifier (DOI) such that it can be cited independently in the future. For instructions see: http://journals.plos.org/plosone/s/submission-guidelines#loc-laboratory-protocols

We look forward to receiving your revised manuscript.

Kind regards,

Andrea Martinuzzi

Academic Editor

PLOS ONE

Reviewers' comments:

Reviewer's Responses to Questions

**Comments to the Author**

1. If the authors have adequately addressed your comments raised in a previous round of review and you feel that this manuscript is now acceptable for publication, you may indicate that here to bypass the “Comments to the Author” section, enter your conflict of interest statement in the “Confidential to Editor” section, and submit your "Accept" recommendation.

Reviewer #2: All comments have been addressed

Reviewer #3: All comments have been addressed

2. Is the manuscript technically sound, and do the data support the conclusions?

Reviewer #2: Yes

Reviewer #3: Yes

3. Has the statistical analysis been performed appropriately and rigorously? 

Reviewer #2: I Don't Know

Reviewer #3: Yes

4. Have the authors made all data underlying the findings in their manuscript fully available?

Reviewer #2: Yes

Reviewer #3: Yes

5. Is the manuscript presented in an intelligible fashion and written in standard English?

Reviewer #2: Yes

Reviewer #3: Yes

6. Review Comments to the Author

Reviewer #2: (No Response)

Reviewer #3: Thank you for addressing my numerous comments from the first revision, they have been sufficiently addressed. It is now clear to me that this paper is not and should not be written exclusively for academic audiences and it will have a major practical impact, thank you for clarifying. This article deserves to be published in PLoS One, but I still have a few minor comments. I apologize that all of these were not brought up in my initial review.

Abstract

Introductory statement. Please consider adding something like “as compared to high and periodized carbohydrate diets.” at the end of your statement to better convey the study purpose.

Line 39. Merely a suggestion, but if you have words available it would help for clarity if you write “After Adapt, athletes resumed HCHO…”.

Line 43-44. Can you add a comparative statement about the trends for HCHO and PCHO groups for whole-body fat oxidation and oxygen cost?

Please have some mention of “rebound” performance for HCHO and PCHO groups in results.

L51. Please change “are robust” to “were repeated”.

Introduction

This reads very well. Thank you for implementing my suggestions.

Methods

All initial queries were sufficiently addressed.

Results

L418. This is a really cool finding and I agree with the suspected muscle glycogen and water content changes! Entirely out of curiosity, has any study directly measured this?

L429. Please ensure that p=0.987 and not p=0.0987

How did you calculate a 95% CI for your effect size? To my knowledge Cohen’s d is the difference in means / SD of the differences and therefore wouldn’t give a 95% CI unless you had an effect size for each subject.

Also, regarding effect sizes, it will be more valuable to report Cohen’s dz as it better represents data in repeated-measure designs vs Cohen’s d. You have many small effect sizes that may be underrepresented compared to Cohen’s dz. Please consider revising.

L449. Please review p value that currently reads p>1.000. Revise all p-values to >0.99 at highest.

Discussion

L839. Please revise “chronic”.

For conclusion, I suggest making the summary of your current findings past tense.

7. PLOS authors have the option to publish the peer review history of their article (what does this mean?). If published, this will include your full peer review and any attached files.

Reviewer #2: No

Reviewer #3: No

---

## [Author Response · Author response to Decision Letter 1]

24 Apr 2020

Thank you for the kind feedback on our revisions. We have addressed the remaining remarks as well as possible

Abstract

Introductory statement. Please consider adding something like “as compared to high and periodized carbohydrate diets.” at the end of your statement to better convey the study purpose.

Line 39. Merely a suggestion, but if you have words available it would help for clarity if you write “After Adapt, athletes resumed HCHO…”.

Line 43-44. Can you add a comparative statement about the trends for HCHO and PCHO groups for whole-body fat oxidation and oxygen cost?

Please have some mention of “rebound” performance for HCHO and PCHO groups in results.

L51. Please change “are robust” to “were repeated”.

We only had 12 words to enable us to make any changes in the abstract. We used these to better clarify the introductory statement, make line 39 clearer and to make the change to L 51.

Introduction

This reads very well. Thank you for implementing my suggestions.

Methods

All initial queries were sufficiently addressed.

Results

L418. This is a really cool finding and I agree with the suspected muscle glycogen and water content changes! Entirely out of curiosity, has any study directly measured this?

Thanks for the feedback. We have a separate paper that covers the changes in body composition and some health parameters (lipids, insulin sensitivity) so we will cover that aspect more clearly in that paper. 

L429. Please ensure that p=0.987 and not p=0.0987

Yes, it is p = 0.987

How did you calculate a 95% CI for your effect size? To my knowledge Cohen’s d is the difference in means / SD of the differences and therefore wouldn’t give a 95% CI unless you had an effect size for each subject.

Also, regarding effect sizes, it will be more valuable to report Cohen’s dz as it better represents data in repeated-measure designs vs Cohen’s d. You have many small effect sizes that may be underrepresented compared to Cohen’s dz. Please consider revising.

We agree that a standard Cohen's d as you describe will not do the repeated measures design justice. However, since we used a linear mixed model estimates (i.e. accounting for the repeated measures) we can calculate a Cohen’s dby including the random effects in the denominator of the effect size (see methods). We have ensured that all are now calculated in this way, and that this has been explained in the methods section;

“Effect sizes based on the classical Cohen’s d were calculated from the linear mixed model estimates, while accounting for the study design by using the square root of the sum of all the variance components (specified random effects and residual error) in the denominator.”

L449. Please review p value that currently reads p>1.000. Revise all p-values to >0.99 at highest.

This has been done

Discussion

L839. Please revise “chronic”.

We have changed this to 3.5 wk

For conclusion, I suggest making the summary of your current findings past tense.

This has been done

---

## [Decision Letter · Decision Letter 2]

19 May 2020

Crisis of confidence averted: Impairment of exercise economy and performance in elite race walkers by ketogenic Low Carbohydrate, High Fat (LCHF) diet is reproducible

PONE-D-19-19340R2

Dear Dr. Burke,

We are pleased to inform you that your manuscript has been judged scientifically suitable for publication and will be formally accepted for publication once it complies with all outstanding technical requirements.

With kind regards,

Andrea Martinuzzi

Academic Editor

PLOS ONE

Additional Editor Comments (optional):

Reviewers' comments:

Reviewer's Responses to Questions

**Comments to the Author**

1. If the authors have adequately addressed your comments raised in a previous round of review and you feel that this manuscript is now acceptable for publication, you may indicate that here to bypass the “Comments to the Author” section, enter your conflict of interest statement in the “Confidential to Editor” section, and submit your "Accept" recommendation.

Reviewer #3: All comments have been addressed

2. Is the manuscript technically sound, and do the data support the conclusions?

Reviewer #3: Yes

3. Has the statistical analysis been performed appropriately and rigorously? 

Reviewer #3: Yes

4. Have the authors made all data underlying the findings in their manuscript fully available?

Reviewer #3: Yes

5. Is the manuscript presented in an intelligible fashion and written in standard English?

Reviewer #3: Yes

6. Review Comments to the Author

Reviewer #3: Thank you for addressing my numerous comments. The work that went into this paper is truly incredible and will be highly impactful.

7. PLOS authors have the option to publish the peer review history of their article (what does this mean?). If published, this will include your full peer review and any attached files.

Reviewer #3: No

---

## [Editor Report · Acceptance letter]

26 May 2020

PONE-D-19-19340R2 

Crisis of confidence averted: Impairment of exercise economy and performance in elite race walkers by ketogenic Low Carbohydrate, High Fat (LCHF) diet is reproducible 

Dear Dr. Burke:

I am pleased to inform you that your manuscript has been deemed suitable for publication in PLOS ONE. Congratulations! Your manuscript is now with our production department. 

With kind regards,

on behalf of

Dr. Andrea Martinuzzi 

Academic Editor

PLOS ONE